

# Towards a GDPR-compliant cloud architecture with data privacy controlled through sticky policies

M. Emilia Cambronero[1], Miguel A. Martínez[1], Luis Llana[2], Ricardo J. Rodríguez[3] and Alejandro Russo[4]

[1] Albacete Research Institute of Informatics, Department of Computer Science, University of Castilla-La Mancha, Albacete, Spain
[2] Universidad Complutense de Madrid, Madrid, Spain
[3] Aragón Institute of Engineering Research, Department of Computer Science and Systems Engineering, University of Zaragoza, Zaragoza, Spain
[4] Chalmers University of Technology, Göteborg, Sweden

## ABSTRACT

Data privacy is one of the biggest challenges facing system architects at the system design stage. Especially when certain laws, such as the General Data Protection Regulation (GDPR), have to be complied with by cloud environments. In this article, we want to help cloud providers comply with the GDPR by proposing a GDPR-compliant cloud architecture. To do this, we use model-driven engineering techniques to design cloud architecture and analyze cloud interactions. In particular, we develop a complete framework, called MDCT, which includes a Unified Modeling Language profile that allows us to define specific cloud scenarios and profile validation to ensure that certain required properties are met. The validation process is implemented through the Object Constraint Language (OCL) rules, which allow us to describe the constraints in these models. To comply with many GDPR articles, the proposed cloud architecture considers data privacy and data tracking, enabling safe and secure data management and tracking in the context of the cloud. For this purpose, sticky policies associated with the data are incorporated to define permission for third parties to access the data and track instances of data access. As a result, a cloud architecture designed with MDCT contains a set of OCL rules to validate it as a GDPR-compliant cloud architecture. Our tool models key GDPR points such as user consent/withdrawal, the purpose of access, and data transparency and auditing, and considers data privacy and data tracking with the help of sticky policies.

# INTRODUCTION

Data privacy was a major concern among scientists before the publication of the General Data Protection Regulation (GDPR) (*Myers & Liskov, 2000*; *Priscakova & Rabova, 2013*). In 2018, the legal text of the GDPR (*General Data Protection Regulation (EU GDPR), 2016*) appeared, which is an extensive document with 99 articles. This regulation directly affects all member states of the European Union (EU), and one of the main novelties with respect

Corresponding authors
M. Emilia Cambronero,
memilia.cambronero@uclm.es
Luis Llana, llana@ucm.es

to previous data privacy legislation is that it also affects any non-EU organization that handles the data of European citizens.

In 2021, the European Commission identified cloud computing as a key vulnerability area (*Euractiv, 2021*). Thus, two codes of conduct for the cloud industry were approved, and these were developed by industry leaders to provide a strategy for GDPR compliance in cloud environments. These codes are focused on increasing trust and transparency in the EU cloud computing market, increasing competition between cloud providers.

The first code of conduct covers Software as a Service (SaaS), and some of its main providers include Alibaba Cloud, Cisco, Dropbox, Google Cloud, Microsoft, and IBM, among others. The second code of conduct covers Infrastructure as a Service (IaaS), and its predominant provider is Amazon Web Services (AWS). These large cloud players do business in a large European cloud market that continues to grow. Therefore, it is vital for cloud providers, both large and small, to update their procedures to comply with the GDPR and be more competitive in this new scenario. In essence, understanding how cloud providers comply with the GDPR represents a key challenge for established and newly emerging providers (*Barati et al., 2019*). According to a study by *Statista (2022)*, many countries have made significant changes to cloud governance after the introduction of the GDPR. For instance, around 63% of the French IT security practitioners estimate that their organization will need major changes in cloud governance after the introduction of the GDPR. This estimate is similar in other countries, such as Germany (57%) or the United Kingdom (56%).

To help companies in this adaptation process, in this article, we use model-driven engineering (MDE) (*Davies et al., 2005*; *Meliá et al., 2016*) to define a Unified Modeling Language (UML) (*Object Management Group (OMG), 2017*) profile for a GDPR-compliant cloud architecture by defining specific stereotypes for this purpose. Our proposal is based on UML and UML profiling techniques, which are well-known software development methodologies in software engineering. These techniques rely heavily on stereotypes, which consist of defining domain-specific types of UML diagram elements. These domain-specific types allow a software designer to create and use UML objects relevant to the problem domain and its terminology (for instance, an actor in a use case diagram or a component in a component diagram that already contain certain attributes, functions, or names relevant to the problem domain). Examples of UML profiles are MARTE (*Object Management Group (OMG), 2011*) (useful for analysis and modeling of embedded and real-time systems), DAM (*Bernardi, Merseguer & Petriu, 2011*) (useful for analysis and modeling of dependability attributes), or SecAM (*Rodríguez, Merseguer & Bernardi, 2010*) (useful for analysis and modeling of security attributes).

Our profile covers both IaaS and SaaS, that is, the cloud infrastructure and the interactions between the different GDPR roles in the cloud when a user stores their data in it. GDPR defines as roles the data subject or user (who owns the data), the third parties (who want access to the data), and the cloud provider (who oversees the user's data). Thus, UML component and sequence diagrams are designed to model the cloud infrastructure and interactions, respectively. We have addressed the main features of the GDPR to ensure the security of user data in our proposed architecture. Some of these features are the

purpose of accessing the data, transparency, audit processes (that is, where user data has been and where it was taken from), and withdrawal of consent to the processing of user data. The profile models are then validated by using Object Constraint Language (OCL) (*Warmer & Kleppe, 2003*) rules to ensure that they comply with certain defined features and constraints. In addition, the proposed cloud architecture allows data tracking and guarantees the privacy of user data. In this regard, data privacy and tracking are controlled using sticky policies (*Pearson & Casassa-Mont, 2011*) associated with the data. The sticky policy allows us to define specific permissions for the data and captures the path followed by data, among other parameters of interest for data tracking.

The UML profile and the OCL rules have been integrated into a complete framework named Modeling Data Cloud Tracking (MDCT). In addition, we have also implemented a software tool that supports MDCT. This tool is publicly available, and its source code has been released under the GNU/GPLv3 license.

In summary, our novel approach allows cloud architectures to track data and guarantee the privacy of user data, complying with many of the GDPR articles. To the best of our knowledge, we are the first to combine model-driven engineering with sticky policies and GDPR in a tool that helps software engineers to adapt GDPR to cloud architectures. In particular, the contributions of this article are the following:

**GDPR-compliant validated UML profile:** We present a GDPR-compliant validated UML profile for cloud architectures, incorporating UML-profiling techniques, UML sequence and component diagrams, and OCL rules for validation. This profile facilitates GDPR compliance to cloud providers. Our profile covers SaaS and IaaS by using UML sequence and component diagrams to model the cloud infrastructure and cloud interactions, respectively. Using the proposed models, cloud providers can provide a data management service that complies with GDPR, and track the data in their systems.

**Data privacy through sticky policies:** Our proposal addresses data privacy by introducing sticky policies, allowing third-party access control through precise data permissions.

**Comprehensive GDPR coverage:** We address the main aspects of the GDPR, including the purpose of data access, consent/withdrawal by the interested party, and transparency and auditing. Additionally, the purpose for which the data is accessed plays a very important role in its treatment. Therefore, in this work, we distinguish between accessing the data for statistical[1] or other purposes.

**Robust data tracking mechanism:** We implement data tracking, monitoring the data's journey and origins. A dedicated log in the controller (cloud provider) and a specific attribute in the sticky policy record information on third-party access.

**Strict OCL rules for validation:** Additionally, we establish strict OCL rules to validate a UML profile targeting cloud service providers. This innovative approach serves as a cornerstone to ensure not only seamless functionality of cloud-based systems, but also critical aspects such as privacy and data tracking. By meticulously defining and applying these OCL rules, our framework sets a new standard for safeguarding sensitive information and enabling effective data tracking in the dynamic cloud computing landscape.

---

[1] Note that "statistical" encompasses a broad range of possible data operations that controllers (cloud providers in this case) themselves must specify. The reason for not distinguishing them is that, regardless of the sub-type of statistical access, no individual can be identified from the resulting data. In GDPR terms, this process is called pseudonymization and is mandatory for any statistical study of personal data.

| Table 1 | List of acronyms. |
|---|---|
| GDPR | General data protection regulation |
| EU | European Union |
| DS | Data subject (physical person who owns the data) |
| SP | Sticky policy |
| TP | Third party |
| SLA | Service level agreement |
| CP | Cloud provider |
| SSM | Stateless storage machine |
| UML | Unified Modeling Language |
| OMG | Object management group |
| OCL | Object constraint language |
| UML-SD | UML sequence diagram |
| MDCT | Modeling data cloud tracking |
| GestF | Third-party business consultancy |
| SB | Santander bank |
| ING | Internationale Nederlanden Groep |
| OCL | Object Constraint Language |
| L-(as in L1, L2, *etc.*) | Location (for data storage) |
| O | Owner(s) of the data (grant permissions over it) |
| P (as in PList) | Principals (entities that access or own data) |
| N-(as in NL, NSP, *etc.*) | New (used as prefix to indicate a modification of a previously referenced variable) |

In order to enhance readability, a list of acronyms is provided in Table 1. The structure of this article is as follows: Related work is discussed in "Literature Review". "Background" introduces the key concepts necessary to understand the rest of this article, such as GDPR, sticky policies, and UML diagrams and profiling. A running example illustrating and motivating the article is presented in "Running Example". The methodology employed in our framework is detailed in "Methodology", while "Modeling Data Tracking in Cloud Systems" describes the UML profile within the MDCT framework. Some of the OCL rules needed to validate the UML models are presented in "Validation and Threat Model", and the tool supporting our framework is introduced in "The MDCT Tool". "Discussion" discusses interesting considerations and common threats to validity. Finally, "Conclusions and Future Work" concludes the article and outlines future lines of work.

## Literature review

In this section, we delve into the existing literature. We categorize our discussion into three main areas: works focused on modeling and validation of the GDPR and data privacy, works dedicated to the GDPR and cloud computing, and works centered around data tracking and the GDPR. Each subsection provides insights into relevant studies,

methodologies, and advancements in these specific domains, laying the foundations for our novel contributions.

### Modeling and validation of GDPR and data privacy

Regarding data privacy modeling, *Basso et al. (2015)* presents a UML profile for privacy-aware applications to build UML models that specify privacy concepts and improve the definition and application of privacy. *Alshammari & Simpson (2018)* also proposes a profile (called APDL) for privacy-aware data to serve as an abstract model for personal data life cycles. In particular, they distinguish between the operations that can be performed on personal data during their lifecycle. While they suggest that the APDL profile could be represented in terms of UML, it does not currently adhere to the UML standard. Notably, these privacy data models do not explicitly consider the GDPR.

Some other works only focus on some specific aspects of the GDPR. For instance, *Mougiakou & Virvou (2017)* propose a model that uses UML use case diagrams, a combination of the GDPR, information privacy, and best practices to examine GDPR requirements using an educational e-platform paradigm, called Law Courses. *Matulevičius et al. (2020)* present a GDPR model and its supporting methods for managing regulatory compliance in business processes. They use component diagrams to model the different aspects of the GDPR, such as consent and data processing. However, they do not model interactions between system roles or consider data tracking.

Likewise, *Politou, Alepis & Patsakis (2018)* assess the impact on personal data protection and privacy of the right to withdraw consent and the right to be forgotten in the GDPR. They consider some existing architectures and technologies to establish whether it is feasible to implement technical practicalities and effectively integrate these new GDPR requirements into current IT infrastructures.

*Torre et al. (2019)* share their experience in creating a UML representation of the GDPR. In particular, they provide several tables with excerpts of the GDPR that are helpful for developers and provide guidelines for creating automated methods to ensure GDPR compliance. However, the authors only use UML package diagrams to design the UML Class Model.

One of the main problems of the application of the GDPR in the field of Information Technology is that it is defined by legal experts, not software or information engineers (*Tamburri, 2020*). Therefore, many works in the literature are devoted to trying to help software engineers in the implementation of the GDPR. Many of these works use modeling techniques, which help with data management and allow software developers to have a global vision of systems.

*Tamburri (2020)* offer a systematic synthesis and discussion of the GDPR by using a mathematical analysis method known as Formal Concept Analysis. Likewise, *Barati, Theodorakopoulos & Rana (2020)* have also formalized the rules and obligations of the GDPR by using timed automata. They check whether the data flow in a business process follows the GDPR guidelines. To do this, they use the UPPAAL tool (*Larsen, Petterson & Yi, 2023*). *Kammüller, Ogunyanwo & Probst (2019)* propose a data label model for GDPR-compliant IoT systems. They apply this model to ensure the protection of patient data in a

health-care system, labeling the data to cover the requirements of the GDPR and presenting several use cases with the labeled data that can be transformed into a formal specification of Object Z. While their label model shares some ideas with sticky policies, it lacks expressive power and focuses on a more restricted problem, maintaining only the owner and a list of authorized actors.

*Vanezi et al. (2020)* only focus on the purpose of data processing. They encode a formal language syntax in a UML-based domain model and present a tool that takes a graphical model definition and then translates it into formal language definitions. *Kaneen & Petrakis (2020)* justify the advisability of GDPR compliance, which is verified in the system design phase by analyzing dependencies between system entities and processes. The authors suggest a series of questions that reflect the GDPR compliance requirements and design class diagrams for these questions. They further generate a series of data reports intended for regulators to evaluate system GDPR compliance during inspections.

In terms of the main differences between these related works and ours, we can highlight the following. Firstly, we focus on GDPR compliance in cloud environments. Secondly, we define a full UML profile to model all interactions in the cloud system and its infrastructure, data tracking, and GDPR compliance, using UML-profile techniques. Thirdly, as modeling techniques, we focus on UML, specifically sequence diagrams and component diagrams, to model the interactions and infrastructure, respectively. Fourthly, the profile models are validated by OCL to ensure compliance with certain restrictions. Finally, we present a tool that supports the entire process.

### Cloud and GDPR

Related works can be classified on whether it targets cloud users or providers. For cloud users or consumers, *Rios et al. (2019)* introduce the DevOps framework. It includes privacy and security controls to ensure transparency for users, third parties, and law enforcement authorities. The framework is based on the risk-driven specification at the design time of privacy and security objectives in the system's service level agreement.

Other works also consider cloud providers. For instance, *Pandit et al. (2018)* define an ontology to represent GDPR. Subsequently, *Elluri & Joshi (2018)* identify the GDPR articles that affect the providers and consumers of cloud services. Then, they develop a more detailed ontology for the obligations of cloud data providers and consumers. In contrast, we have focused on tracking user data to ensure the rights of the data subject. For this reason, we have also considered Chapter III of the GDPR (articles 12 to 23), which was not contemplated in *Elluri & Joshi (2018)*.

*Razavisousan & Joshi (2021)* develop a methodology called Textual Fuzzy Interpretive Structural Modeling, which analyzes large textual data sets to identify driving and dependent factors in the dataset. They identify the critical factors in the GDPR and compare them with various Cloud Service privacy policies. Their results show different factors that stand out in the GDPR and other privacy policies of publicly available services. The authors state that this methodology can be used by both service providers and consumers to analyze how closely a service's privacy policy aligns with the GDPR. The

focus of their work is different from ours, as we propose a cloud architecture for cloud providers which ensures GDPR compliance and includes privacy policies *via* sticky policies associated with user data.

As for those works that are more oriented towards cloud providers, *Georgiopoulou, Makri & Lambrinoudakis (2020)* identify the requirements and appropriate countermeasures for GDPR compliance in cloud environments. They describe the GDPR-related features, requirements, and measures that follow the cloud architecture. *Shastri, Wasserman & Chidambaram (2019)* examine how the design, architecture, and operation of modern cloud-scale systems conflict with the GDPR. They illustrate these conflicts through what they call GDPR anti-patterns. They then present six system design and operation anti-patterns, which are effective in their context but violate the GDPR rights and receipts. They propose that cloud designers examine their systems for these anti-patterns and remove them. This work focuses on studying and avoiding these specific patterns, but they do not propose a GDPR-compliant cloud architecture for a cloud provider.

*Mohammadi et al. (2018)* define a comprehensive architecture for runtime data protection in the cloud. They identify five important actors and entities in the GDPR: data subject, data controller, sensitive data, application, and infrastructure. They also derive nine requirements from the architecture and use UML to design and validate this architecture. This work is focused on data security rather than how data permissions are granted by verifying third-party access. Unlike their work, we detail the interaction of third-party software applications that want to access the data, and how their permissions are checked.

*Fan et al. (2019)* and *Chadwick et al. (2020)* emphasize user-centered data sharing, addressing data sharing agreements and employing privacy-preserving methods. While these aspects are important in the context of GDPR, none of these works specifically addresses GDPR compliance as we do in this work. Instead, our focus is on cloud providers and GDPR, ensuring that data tracking and access are restricted to authorized entities.

*Zhou, Barati & Shafiq (2023)* propose a domain model of the accountability principle in the GDPR. The authors use a blockchain-based technique to provide data immutability and integrity for cloud providers' data processing activities. In contrast, we provide a UML profile focused on data tracking that ensures GDPR compliance by design.

In summary, none of the cited works model the cloud system using UML or incorporate data tracking as we do in our work. Moreover, our primary focus is on aiding cloud providers in designing GDPR-compliant cloud architectures.

### Data tracking and GDPR

*Gjermundrød, Dionysiou & Costa (2016)* present a GDPR-compliant tool that covers data transparency and treatability, called privacyTracker. They implement data portability and the right to erasure as contained in the GDPR rights. This framework empowers consumers with the appropriate controls to track the disclosure of data collected by companies and assess the integrity of these multi-handled data. In this article, we not only

**Table 2  GDPR articles and recitals considered in this work.**

| GDPR | Article description |
|---|---|
| Article 4 | Definitions |
| Article 5 | Principles relating to processing of personal data |
| Article 6 | Lawfulness of processing |
| Article 7 | Conditions for consent |
| Article 8 | Conditions applicable to child's consent in relation to information society services |
| Article 9 | Processing of special categories of personal data |
| Article 12 | Transparent information, communication, and modalities for the exercise of the rights of the data subject |
| Article 13 | Information to be provided where personal data are collected from the data subject |
| Article 14 | Information to be provided where personal data have not been obtained from the data subject |
| Article 15 | Right of access by the data subject |
| Article 16 | Right to rectification |
| Article 17 | Right to erasure ("right to be forgotten") |
| Article 21 | Right to object |
| Article 22 | Automated individual decision-making, including profiling |
| Article 24 | Responsibility of the controller |
| Article 25 | Data protection by design and by default |
| Article 28 | Processor |
| Article 29 | Processing under the authority of the controller or processor |
| Article 33 | Notification of a personal data breach to the supervisory authority |
| Article 34 | Communication of a personal data breach to the data subject |
| Article 55 | Competence |
| Recital 44 | Performance of a contract |
| Recital 109 | Standard data protection clauses |

consider data tracking and the rights of data portability and the right to erasure but many other GDPR rights too. All the GDPR articles considered in this work are summarized in Table 2.

With regards to works that are more focused on data tracking and the GDPR, it is worth mentioning the following. *Barati et al. (2019)*, *Barati & Rana (2020)* focus on the issue of GDPR compliance using Blockchain technology. The GDPR compliance of the operations performed is verified using smart contracts. Their work is based on a voting mechanism of the actors to reach a GDPR compliance verdict. If there is a violation, the actor who committed it is informed. In our case, the service provider is responsible for guaranteeing correct access to the data. Therefore, we have a log that saves all data accesses, and this log contains the actions on the user's data. In this way, users can be informed about the use of their data (GDPR Articles 12, 13, and 14). Subsequently, *Barati et al. (2020)* propose three smart contracts to support the automated verification of GDPR operations performed on user data on smart devices. They present a formal model to support GDPR compliance for these devices. The privacy requirements of such applications are related to the GDPR obligations of the device.

### GDPR-compliance assistant tools

Some tools help developers comply with the GDPR. GDPRValidator (*Cambronero et al., 2022*) helps small and medium-sized enterprises that have migrated their services to achieve GDPR compliance. PADRES (*Pereira, Crocker & Leithardt, 2022*) is a tool aimed at web developers, which is organized by principles in the form of a checklist and questionnaire. They also integrate other open-source tools to scan the web project. RuleKeeper (*Ferreira et al., 2023*) is another tool to help web developers. In this tool, web developers specify a GDPR manifest that is automatically incorporated in the web application and is then enforced using static code analysis and runtime mechanisms. In contrast, our tool is a modeling tool aimed at cloud providers to develop systems that comply with GDPR by design.

## BACKGROUND

This section covers some key concepts necessary to understand the rest of this article. We first explain what the European General Data Protection Regulation is, and then we discuss sticky policies. Finally, we briefly describe the Unified Modeling Language and the Object Constraint Language.

### The General Data Protection Regulation (GDPR)

The General Data Protection Regulation (GDPR) (*General Data Protection Regulation (EU GDPR), 2016*) came into force on May 25, 2018, as a way to harmonize data protection rules within EU member states. The GDPR was adopted in 2016 to replace the Data Protection Directive, which was born in 1995 out of a need to align data protection standards within its EU member states to facilitate internal and cross-border EU data transfer.

The GDPR is a regulation, which means that it applies directly to its recipients, and no further transpositions are required, as in the case of the Data Protection Directive. In addition to equalizing the data protection rules, the GDPR was introduced to generate greater legal certainty and eliminate potential obstacles to the free flow of personal data, raising the bar for the privacy of the affected persons.

The GDPR applies to any processing of personal data (or personal data sets), whether the processing is carried out, in whole or in part, by automated means (*General Data Protection Regulation (EU GDPR), 2016*). Anyone who processes or controls the processing of personal data is subject to the GDPR. There are different actors in the GDPR: *data subjects*, who are the people whose data is processed (for example, customers or site visitors); *controllers*, which can be a natural or legal person, public authority, agency, or other body that determines the purposes and means of the processing of personal data; and *processors*, who are a natural or legal person, public authority, agency, or other body that processes personal data on behalf of the controller. The data can be processed within its organization (that is, the controller and the processor are the same) or delegated to an external organization.

Any individual benefits from the GDPR, which also provides specific protection to minors. In contrast, legal entities do not benefit from protection under the GDPR,

regardless of their legal form. The GDPR applies when the processing of personal data takes place within the EU or it involves data obtained form European citizens outside of the EU.

There are various implications of the GDPR for organizations and entities (*European Comission, 2016*). One of the most relevant implications is fair data processing, which means that organizations and entities must process personal data in a legal, fair, and transparent manner. In addition, they must demonstrate that they are GDPR-compliant (*accountability*) and put in place the necessary technical and organizational measures to guarantee the protection of personal data. The GDPR also establishes the *purpose limitation*, which means that personal data is collected for specified, explicit, and legitimate purposes and that no further processing is performed in a manner incompatible with those purposes.

The GDPR incorporates a systems engineering approach called *privacy by design*. This approach is based on seven fundamental principles that aim to proactively integrate data protection into the design of new products and systems. These principles are as follows (*Langheinrich, 2001*; *Cavoukian, 2009*): (i) proactive not reactive; preventive, not remedial; (ii) privacy as the default setting; (iii) privacy embedded into the design; (iv) full functionality—positive-sum, not zero-sum; (v) end-to-end security—full life-cycle protection; (vi) visibility and transparency—keep it open; (vii) respect for user privacy— keep it user-centric.

In the event of a data breach, organizations and entities under the GDPR must inform the data protection authorities within the next 72 h after they become aware of the personal data breach, and inform their users promptly. Infractions of different types (less serious or serious) are applied to organizations and entities if the notification is not made on time or the data breach was caused by the negligence of the controller or the processor of personal data.

Another important aspect of the GDPR is the empowerment of data subjects with certain rights to help data subjects in being assured of the protection and privacy of their personal data (*General Data Protection Regulation (EU GDPR), 2016*). These data subject rights are as follows: right to information, right of access, right to rectification, right to erasure, right to restriction of processing, right to data portability, right to object, and the right to avoid automated decision-making.

## Sticky policies

A sticky policy defines a set of conditions and restrictions attached to data that describe how the data should be treated or, where applicable, transmitted between parties (*Pearson & Casassa-Mont, 2011*). The use of sticky policies facilitates compliance with, and the application of, data policy requirements, since it allows strict control of the data life-cycle in order to guarantee its privacy and the application of specific regulations on the use, access, and disclosure of personal data.

Sticky policies enhance data owners' control over their data. In particular, machine-readable policies are directly attached to the data, and they are called sticky since they travel along with the data as it travels across multiple administrative domains. These

policies make it possible to regulate how data can be accessed and used throughout its life cycle, helping to ensure that access control decisions and policy applications can be carried out in a distributed manner.

This paradigm was initially proposed by *Karjoth, Schunter & Waidner (2002)* to formalize applicable regulations and associate them with collected data, thereby supporting the identification of applicable regulations and privacy expectations for all personal data in a company. Pearson and Mont were early adopters of sticky policies in the context of the EnCoRe project (*Pearson & Casassa-Mont, 2011*), which provided mechanisms for users to define and change consent policies, as well as to enforce these policies throughout the entire data life-cycle.

Among other things, a sticky policy can define who owns the data, the content of the data (it may be encrypted), the use to be made of the data (*e.g.*, for statistical analysis, transaction processing, targeted marketing), who can access the data, the maximum duration of the data, as well as other specific obligations and restrictions for the parties involved.

## Unified Modeling Language (UML) sequence diagrams and combined fragments, UML profiles, and the object constraint language (OCL)

The Unified Modeling Language (UML) (*Object Management Group (OMG), 2017*) is a modeling graphical language commonly used in the industry for specification, design, visualization, and documentation of software systems. UML includes several diagram notations for modeling different aspects of software systems, addressing its structural, behavioral, and deployment aspects.

A UML sequence diagram (UML-SD) is a behavioral diagram of the software system that illustrates the sequence of messages passed between system participants (users or system elements) in an interaction. Therefore, a sequence diagram consists of a group of entities or roles that interact in a system, represented by vertical lifelines, and horizontal arrows that represent the messages that they exchange during the interaction over time. In a UML sequence diagram, a lifeline represents an individual participant, object, or entity involved in an interaction or collaboration. It is depicted as a vertical dotted line, headed by a rectangle or cube with the name of the object it represents, and it is used to show the chronological order of interactions between objects in the system.

In a UML-SD, a combined fragment reflects one or more aspects of interaction (called interaction operands) controlled by an interaction operator. The combined fragments are represented by a rectangle and contain the conditional structures that affect the flow of messages (the interaction operands). A combined fragment separates the contained interaction operands with a dashed horizontal line between each operator.

The combined fragment type is determined by the interaction operator. For instance, the operator *loop* allows the software designers to express interaction loops, while the operator *alt* allows them to express alternative flows of messages. The operator *opt* allows the modeling of an *if-then* structure. Finally, a combined fragment can also contain nested combined fragments or interaction uses (operator *ref*), whose main goal is to reference

other interactions in a UML sequence diagram, and they make it possible to simplify large and complex sequence diagrams.

UML can be adapted for analysis purposes through profiles, by using a UML tool called UML profiling. A UML profile is an extension of the UML standard language with specific elements that correspond to the same domain. For instance, the MARTE (*Object Management Group (OMG), 2011*) profile has enabled UML to specify and analyze embedded and real-time systems. Likewise, the performance and schedulability sub-profiles of MARTE have proved useful for the modeling and analysis of a wide range of application domains, apart from real-time systems.

The Object Constraint Language (OCL) is part of the UML set of modeling notations (*Warmer & Kleppe, 2003*). OCL provides a precise textual language for model validation by expressing constraints that cannot be shown diagrammatically in UML. For instance, OCL constraints can be used to specify that a certain attribute must be unique within a class, or that a method must only be called if a particular precondition is met.

By using OCL, software developers can describe constraints and expressions on UML models that must hold on to the UML model elements. In practice, OCL constraints are often used to complement the UML modeling process, as they can help identify potential bugs early in the development cycle. When validating a UML model using OCL, it is possible to catch errors or inconsistencies in the model and correct them before the implementation phase begins, thus improving the quality of the resulting software system. Hence, OCL is a powerful tool for validating UML models and ensuring their correctness and completeness (*Oestereich, 2002*; *Völter et al., 2006*).

## Stateless machines

Stateless machines (*Sbarski & Kroonenburg, 2017*; *Villamizar et al., 2016*) are software components or systems that operate without maintaining session state information for individual users or clients. They rely on external sources to obtain necessary state information and comply with rigorous security measures to ensure data reliability and integrity. In short, when a state machine is launched, it loads data from a data store, and computes some results which are then stored or sent back to the data processing pipeline. Lambdas AWS (https://aws.amazon.com/lambda/) is an example of such a computational model.

Stateless machines play a crucial role in contemporary software design, providing several advantages in scalability, fault tolerance, performance, and streamlining system deployment and maintenance. In the context of GDPR compliance, they help improve security by mitigating the risks associated with data leakage and unauthorized access that can arise from storing user session data. Its importance is particularly pronounced in distributed and cloud-based systems, where reliability and efficiency are paramount attributes. Maintaining meticulous design principles and implementing robust security practices is imperative to ensuring the trustworthiness of external state information. Stateless machines present compelling benefits particularly in industries where secure data management is of utmost importance, such as banking.

# RUNNING EXAMPLE

In this section, a running example is presented to illustrate the usefulness of our proposed GDPR-compliant cloud architecture. It consists of a business consultancy that runs several applications in the cloud for which it must read and write a variety of sensitive data in the context of GDPR. Note that special attention is paid to the sticky policies associated with this data and how to set the corresponding sticky policy when new data is generated, as a result of combining or aggregating data, to ensure the privacy of the new data. We also address data tracking. For this purpose, a specific field of the sticky policy, called *accessHistory*, is defined to keep track of who is accessing the data and for what purpose.

The roles that interact in the cloud system are the following: the owner of the data or user; the cloud provider, which acts as the data controller; the business consultancy (called *GestF*), which is a third party that wishes to access the user data to perform certain operations on them; and a processor (named *SSMProcessor*), which represents a stateless storage machine in the cloud, where the processing is performed on behalf of a controller. In this example, *GestF* can access the data for two different purposes: to provide customers with tax returns (tax purposes), or to calculate certain population statistics (statistical purposes).

Figure 1 shows the interactions between roles in the cloud using a part of our UML profile, which is described in more detail in "Modeling Data Tracking in Cloud Systems". The first two messages, namely *SPDataSubject* (SP refers to Sticky Policy) and *SLA* (Service Level Agreement), correspond to the contracts signed between the owner of the data (data subject (DS)) and the controller (cloud provider), and between the controller and the processor (*SSMProcessor*), respectively, according to GDPR, Article 28, Recitals 44 and 109. The message *sendData* models the sending of data from the *user* to the controller and from the controller to the processor. It also specifies the data retention period (in this case, 180 days). The message *info* models the fact that the data controller must inform the user about who is responsible for processing its data, and the retention time once the contracts are signed, according to the GDPR. Therefore, during these 180 days (*time ≤ 180 days* condition), *GestF* can express its desire to access this data (*alt[GestF wants to access Data]*), but *GestF* needs the data owner's consent for data access. The messages *consent*, *askAuthentication*, and *GestF* are used for this purpose. In the event that the interested party or user consents to data access (*alt[User consents]*), the message *ok* is sent, and *GestF*'s access to information is added in the controller log (*AccessLog*), *via* the *adding access information in AccessLog* action. The *permission* message is then sent.

Once *GestF* has permission to access the data, it can access it for the two different purposes mentioned: statistical or tax. The main difference between these purposes focuses on the resulting privacy restrictions (sticky policy) for the new data obtained from the calculations performed, which generally involve a combination of different data. For tax purposes, the resulting data owners are all the owners of each combined set of data, while the permissions are limited to the most restrictive for each of them.

Note that we use a special type of *purpose* called *statistical*. In this case, the results of computing the data with this purpose turn out to be new data where no individual can be

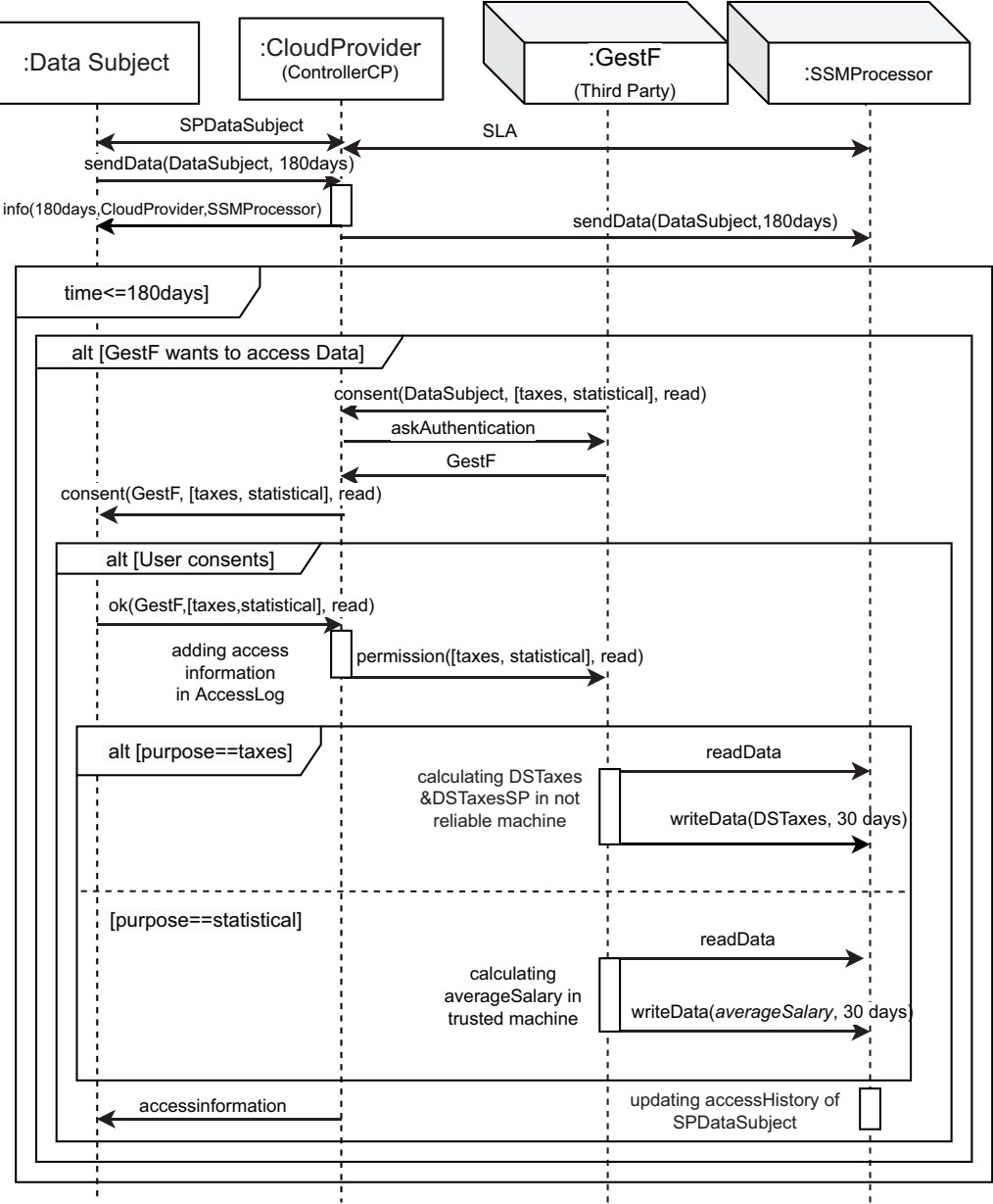

**Figure 1 Running example: UML sequence diagram representing the interaction of *GestF* in the cloud.**

identified. In order to enforce this type of computation, our architecture considers *trusted stateless* machines that guarantee that such statistics are generated using privacy-enhanced technology, such as differential privacy or k-anonymity. In other words, we call trusted stateless machines to those stateless machines that leak information in a controlled manner. In this architecture, we have considered that providers offer their services through stateless machines, which do not store user-session information. These kinds of machines are usually less costly than using a stateful one for a similar purpose as the maintaining entities do not have to manage the resident memory. This presents an interesting offer for

providers as, in the case they want to store session information, they must do so through software cookies, which are regulated in the GDPR. In this case, *GestF* reads the data of the interested party and calculates the average of the salaries of the employees, which is aggregated data (we assume that the data about the other employees have been previously read). This calculation is performed on a *trusted* machine, which our cloud architecture provides specifically for this purpose. The sticky policy of the data obtained will be different from that obtained for non-statistical purposes since the person who generates the data is the data owner, as explained in "Final Sticky Policies".

If the purpose is not statistical, unreliable cloud machines (*regular* machines) are used. If *GestF* accesses the cloud to calculate the user's taxes (*alt[purpose==taxes]*), it reads its data and performs the tax calculation on a *regular* machine. To do this, it combines *GestF* data with the user data and writes the new data (taxes) to the storage machine for 30 days (*readData* and *writeData* messages). The resulting sticky policy for these newly obtained data is explained in "Final Sticky Policies".

## Initial sticky policies

We consider five main fields in the sticky policies: *permission*, *owner*, *purpose*, *controller*, and *accessHistory*. The *permission* field defines the access to the data. Permissions are defined as DC labels (*Stefan et al., 2012*), which are tuples $\langle S, I \rangle$ where $S$ and $I$ are conjunctive normal forms on entities without negative literals. $S$ specifies the entities whose consent is required to access the data, while $I$ specifies the entities that have created the data and may modify it. A more detailed explanation is given in "Combination and Data Aggregation".

The *owner* field defines the owner of the data, while the *purpose* field defines a list of possible access purposes (in this running example, statistical or tax purposes, as explained above). The *controller* defines the data controller according to the GDPR, in this case it will be the cloud provider. Finally, the *accessHistory* field allows us to track this data, that is, save all the entities that have accessed the data and the purpose of the access.

The sticky policies for the different data used in this example are as follows. For the data of the interested party, data subject (*DS*) or user:

```
DS_SP = {
  {permission: ⟨DS,DS⟩},
  {owner: DS},
  {purpose: taxes, statistical},
  {controller: ControllerCP},
  {accessHistory:
    [(SB, statistical, read),
    (ING, statistical, read)]}
}
```

Therefore, *DS* is the only one that can grant access to the data and can also *write* the data. The owner of this data is the *DS*. The list of purposes has two types: *statistical* and *tax*. Finally, the *accessHistory* field allows us to track this data. We assume two new entities

(*SB* and *ING*), representing banks. These entities are allowed to *read* the data for *statistical* purposes.

As can be seen, *GestF* does not have permission to access the *DS* data in the initial sticky policy, so it needs to request consent from the controller, who in turn will ask the data subject (*DS*) for consent. As shown in Fig. 1, consent is given to *GestF* to *read* the data from *DS*, who first reads the data. Therefore, the access history in the *DS* data must be updated to reflect this access:

```
DS_SP = {
  {permission: ⟨DS,DS⟩},
  {owner: DS},
  {purpose: taxes, statistical},
  {controller: ControllerCP},
  {accessHistory:
   [(SB, statistical, read),
   (ING, statistical, read),
   (GestF, taxes, read),
   (GestF, statistical, read)]}
}
```

Likewise, *GestF* data has the following sticky policy:

```
GestFDataSP = {
  {permission: ⟨GestF,GestF⟩},
  {owner: GestF},
  {purpose: taxes, statistical},
  {controller: ControllerCP},
  {accessHistory: []}
 }
```

This sticky policy means that *GestF* must grant access to the data and can *write* its data, the owner is *GestF*, and the list of purposes has both types: *statistical* and *tax* purposes. Unlike before, the *accessHistory* field is empty, which means that no one has accessed *GestF*'s data yet. Eventually, *GestF* calculates the *DS*' taxes and stores them for 30 days (*writeData(DataSubjectTaxes, 30 days)* message). This creates new data whose sticky policy is a combination of data from *GestF* and data from *DS*.

## Final sticky policies

The final sticky policies obtained for the new data generated because of the behavior shown in Fig. 1 depend on purpose of the access (note that the controller remains the same):

● *Statistical purpose.* In this scenario, *GestF* executes a statistical application in the cloud to compute the average salary for its employees (in this example, *DS* and *DS1*). As *GestF* already possesses *statistical* access to *DS*'s data, the controller is not required to seek the user's permission for access. The application runs on a *trusted* machine, as detailed earlier. Following the computation, the anonymized aggregated data is generated and written to

the storage machine (*writeData(averageSalary, 30 days)* message) with the ensuing sticky policy:

```
averageSalarySP={
  {permission: ⟨GestF,GestF⟩},
  {owner: GestF},
  {purpose: statistical},
  {controller: ControllerCP},
  {accessHistory: []}
```

As can be seen, the sticky policy of the new data is different since we consider that in the statistical case the owner of the new combined data is the one who generates this new data (in this case, *GestF*), and then decides on its permissions. Here, the *purpose* is statistical only, and the *accessHistory* only considers the access of *GestF*, since this data will be used solely in the interest of *GestF*.

• *Tax purpose.* In this case, the *GestF* is running a tax application on a regular cloud machine and combining its data with the data from *DS* (*SPGestF* ⊔ *DS_SP*) to calculate the *DS*'s taxes (*DSTaxesSP*), where the operator ⊔ is ∧ for the entities required to give consent for *read* operations, and ⊔ is ∨ for *write* operations. So, the new sticky policy for the new data (tax data) is as follows:

```
DSTaxesSP=
  {permission: ⟨DS ∧GestF,DS ∨ GestF⟩},
  {owner: DS ∧ GestF},
  {purpose:taxes, statistical},
  {controller: ControllerCP},
  {accessHistory: []}
```

Note that, for tax purposes, each field is generated according to the following rules:

– The *permission* field is obtained from the most restrictive combination of the permissions of *GestF* and DS. That is, the DS_SP permissions are ⟨*DS*, *DS*⟩, while the *Gest* SP permissions are ⟨*GestF*, *GestF*⟩. Therefore, the resulting permission is: ⟨*DS*, *DS*⟩ ⊔ ⟨ *GestF*, *GestF*⟩ = ⟨*DS* ∧ *GestF*, *DS* ∨ *GestF*⟩.

– The *owner* field contains all the owners of the combined data; in this case, *DS* and *GestF*.

– The *accessHistory* is empty because it is new data, and no one has requested access to it yet.

## METHODOLOGY

This section presents the methodology followed by our proposed Modeling Data Cloud Tracking (MDCT) framework. The main objective of our framework is to define recommendations that allow cloud providers to create a stateless computing architecture in the cloud that complies with the GDPR and guarantees the privacy of cloud users. For

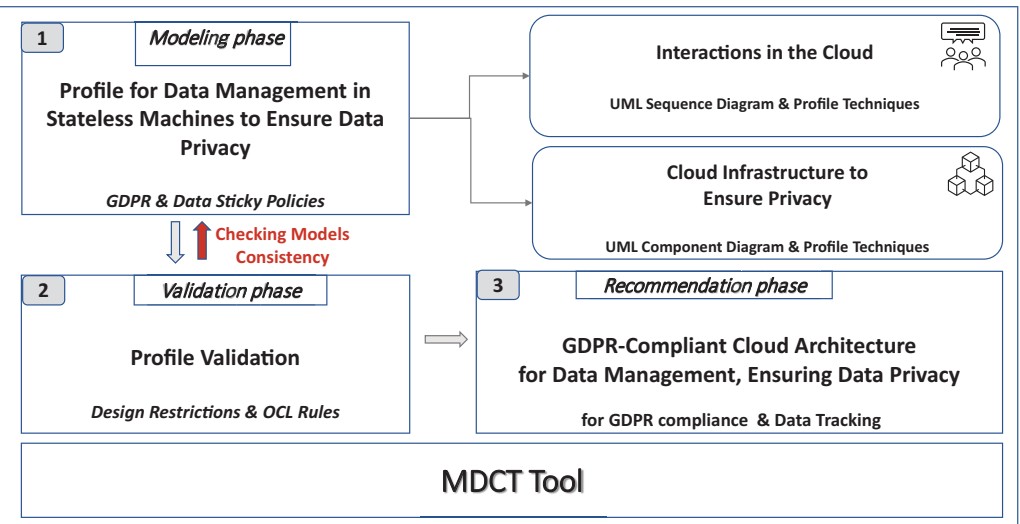

**Figure 2  Modeling data cloud tracking (MDCT) framework proposal.**

this purpose, we focus on designing a GDPR-compliant cloud architecture that uses sticky policies to ensure data privacy. In addition, the use of sticky policies allows our framework to track user data throughout their entire life-cycle.

Figure 2 describes the different phases of our framework, which are as follows:

Phase 1. Modeling phase. The UML profile, named Model4_DataCTrack, is modeled to design the proposed GDPR-compliant cloud stateless computing architecture. For this purpose, we have had the support of expert GDPR consultants. Model4_DataCTrack uses two types of UML diagrams, namely sequence and component diagrams, which allow us to define the interaction between the roles in the system and the cloud architecture infrastructure, respectively. We consider several parameterized sequence diagrams that define the behavior of GDPR roles and third parties when accessing and managing sensitive data in the cloud. Next, the specific configuration of the cloud infrastructure is established by setting the corresponding parameters in the component diagram. This infrastructure considers the sticky policies associated with the data to ensure data privacy. Then, UML profiling techniques (*Malavolta, Muccini & Sebastiani, 2015*) are also used to model the specific stereotypes needed. The GDPR articles considered are specifically indicated in the description of the models. More details on this matter are given in "Modeling Data Tracking in Cloud Systems".

Phase 2. Validation phase. The model generated must comply with certain properties that are validated in this phase. In this regard, we define a set of OCL (*Object Management Group (OMG), 2014*) rules, which allows us to detect errors and warnings in the model. For instance, if the action to perform on the data is not of the allowed action type, an error is detected. If errors are detected, they must be corrected, and we return to the previous phase (Phase 1. Modeling phase) to correct them. After that, the model must be validated again.

The validation of our profile model is described in detail in "Validation and Threat Model".

Phase 3. Recommendation phase. After model validation, this phase allows cloud providers to finalize a stateless computing architecture configuration in the cloud. This configuration is GDPR-compliant, ensures user privacy, and allows data tracking.

## MODELING DATA TRACKING IN CLOUD SYSTEMS

This section resulting describes Model4_DataCTrack and its validation in detail. We first look at the interaction model, showing the UML sequence diagrams that allow us to model the interaction between the different roles in the system. We then introduce the rules for generating the new sticky policy for new data and for the aggregation or combination of this data, when data has been accessed for statistical or other purposes, respectively. Finally, we present the profile stereotypes and system infrastructure using UML component diagrams.

In what follows, we adhere to terminology expressed in Article 4 GDPR "Definitions" (*General Data Protection Regulation (EU GDPR), 2016*) for the main definitions and concepts used in our model. The specific articles and recitals of the GDPR considered for this work are summarized in Table 2. We explicitly mention them in the description of our profile.

### Interaction model

This section describes the UML sequence diagrams that model the interaction between the different roles in our proposed stateless cloud architecture. Regarding the roles, we have defined the following: the *user* (also called *data subject*), the *ControllerCP*[2] (controller), the stateless computer applications (StatelessAppTP[3]), which want access to the data, and finally, the SSMProcessor[4] (processor), which is the machine on which the data is stored.

User data is considered sensitive information to be stored and processed in the system. Therefore, ControllerCP is responsible for implementing appropriate technical and organizational measures to guarantee and be able to demonstrate secure access to data (Art. 24 and 25 GDPR (*General Data Protection Regulation (EU GDPR), 2016*)). It is then responsible for monitoring the application of GDPR to protect the fundamental rights and freedoms of natural users with respect to data processing, and for facilitating the free flow of sensitive data within the EU. The SSMProcessor is responsible for data processing (Article 28 GDPR; *General Data Protection Regulation (EU GDPR), 2016*). In this cloud environment, the stateless storage machine acts as the data processor, as it stores the data and is responsible for data processing. In accordance with Article 29 GDPR (*General Data Protection Regulation (EU GDPR), 2016*), the processor and any person acting under the authority of the controller or the processor, who has access to personal data, shall not process this data except on the instructions of the controller, unless required by the law of the Union or Member State.

Figures 3–6 show the interaction between the different roles using UML sequence diagrams.

[2] CP is the abbreviation for Cloud Provider.

[3] TP is the abbreviation for Third Party.

[4] SSM stands for Stateless Storage Machine.

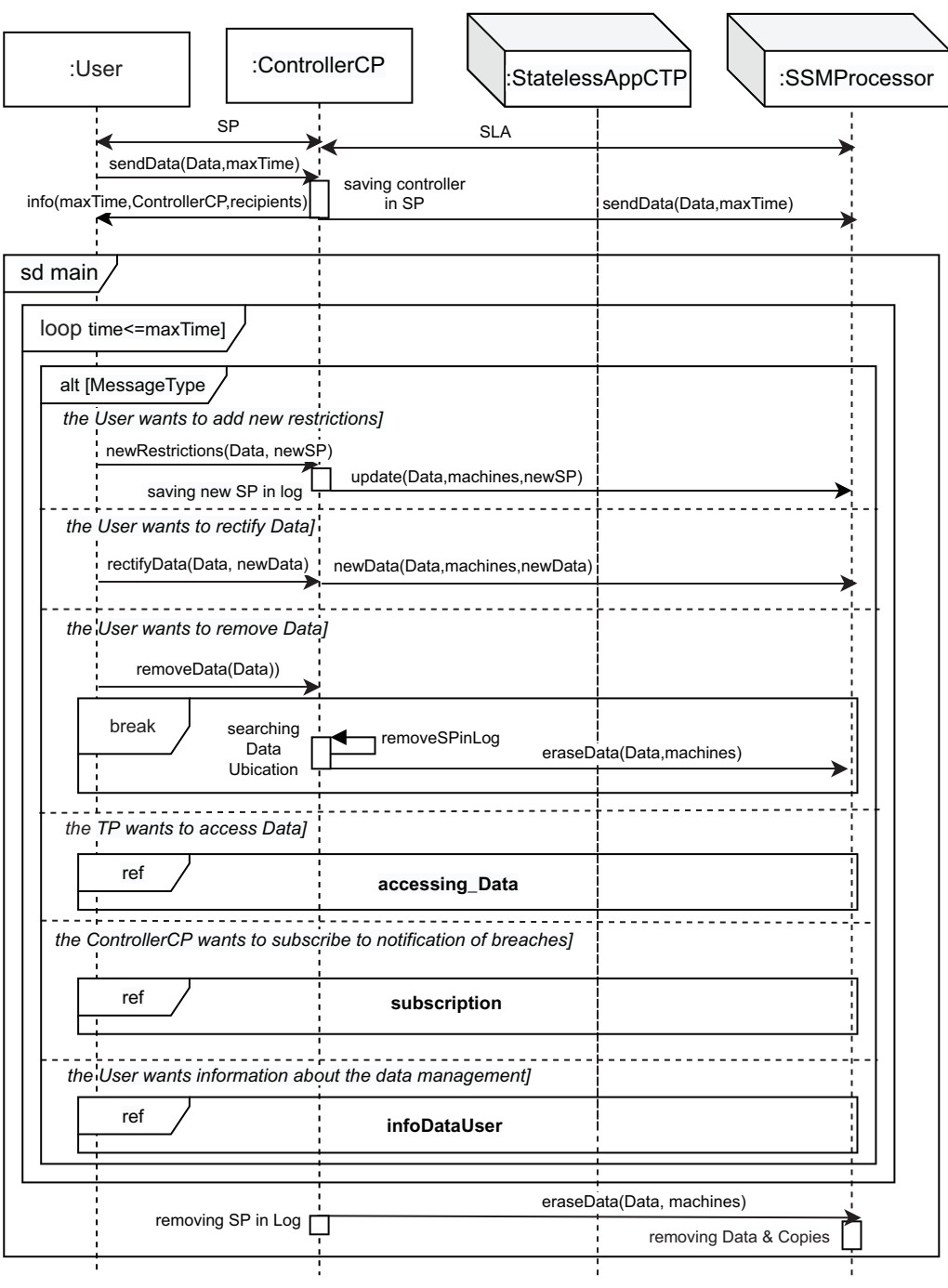

**Figure 3** Main SD: main interaction diagram in the cloud.

5 It is important to remark that the treatment of data belonging to minors only differs from other categories of data in the collection process. In these cases, controllers must make the information about the processing more accessible and clear, and require the consent of the certified legal guardians of the individual. This is specified in Article 8 and recital 38 GDPR (*General Data Protection Regulation (EU GDPR), 2016*).

Figure 3 shows the main sequence diagram, in which we capture the interactions of the different roles when a user interacts with the cloud system and sends personal data to it. As personal data, we also consider data of special categories (Article 9 GDPR (*General Data Protection Regulation (EU GDPR), 2016*)) or children's data[5]. However, in the proposed architecture we do not consider personal data relating to criminal convictions and offenses

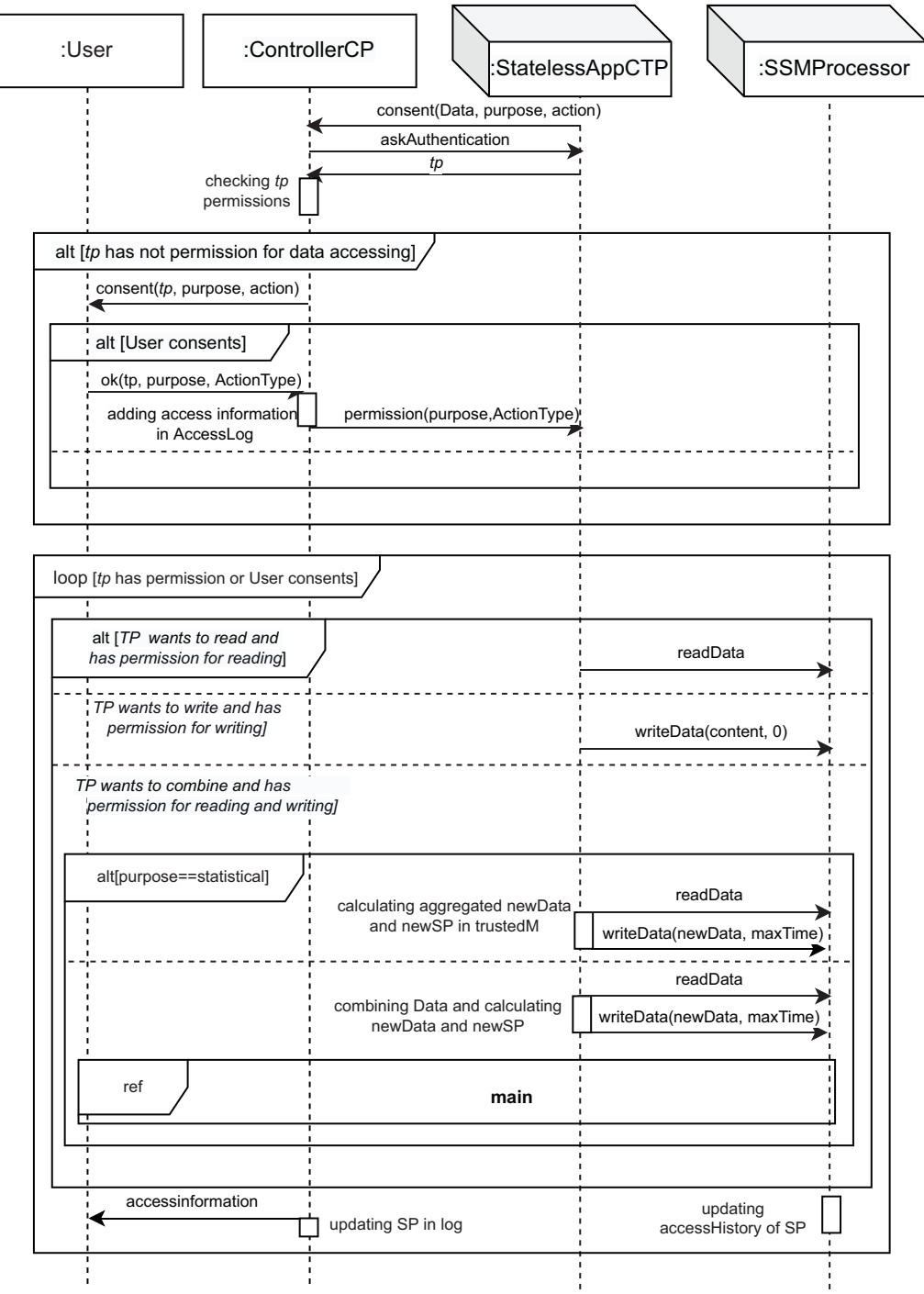

**Figure 4  Accessing_Data SD: third parties accessing data.**

(Article 10 GDPR (*General Data Protection Regulation (EU GDPR), 2016*)). Initially, the user signs a contract (represented by the Sticky Policy (SP) message) with the controller to establish it as the controller of data processing and guarantee the principles relating to the processing of personal data (Article 5 GDPR (*General Data Protection Regulation*

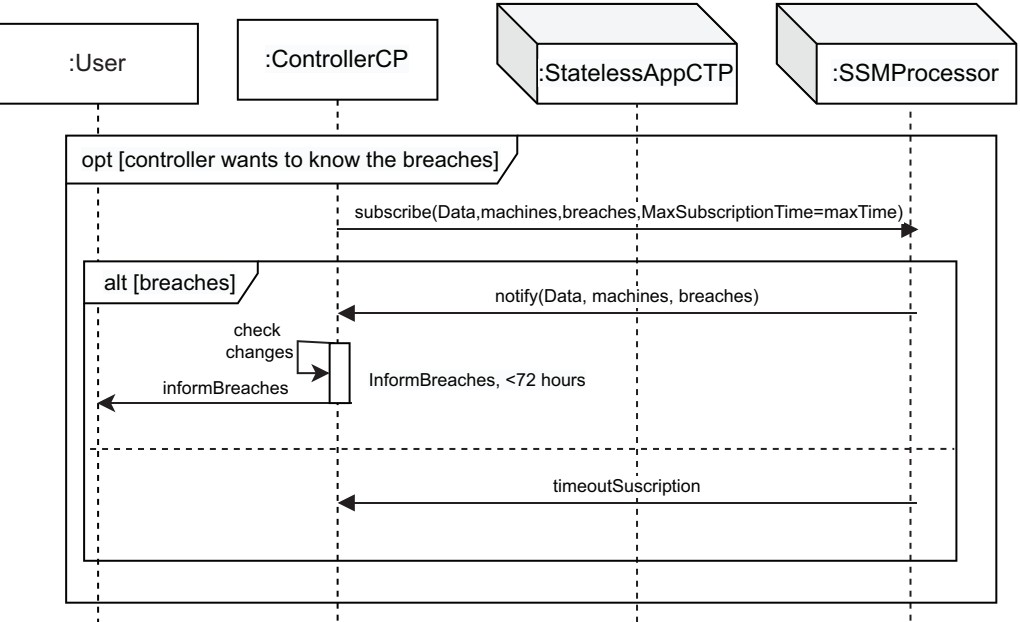

**Figure 5 Subscription SD: controller subscription to be notified when data changes.**

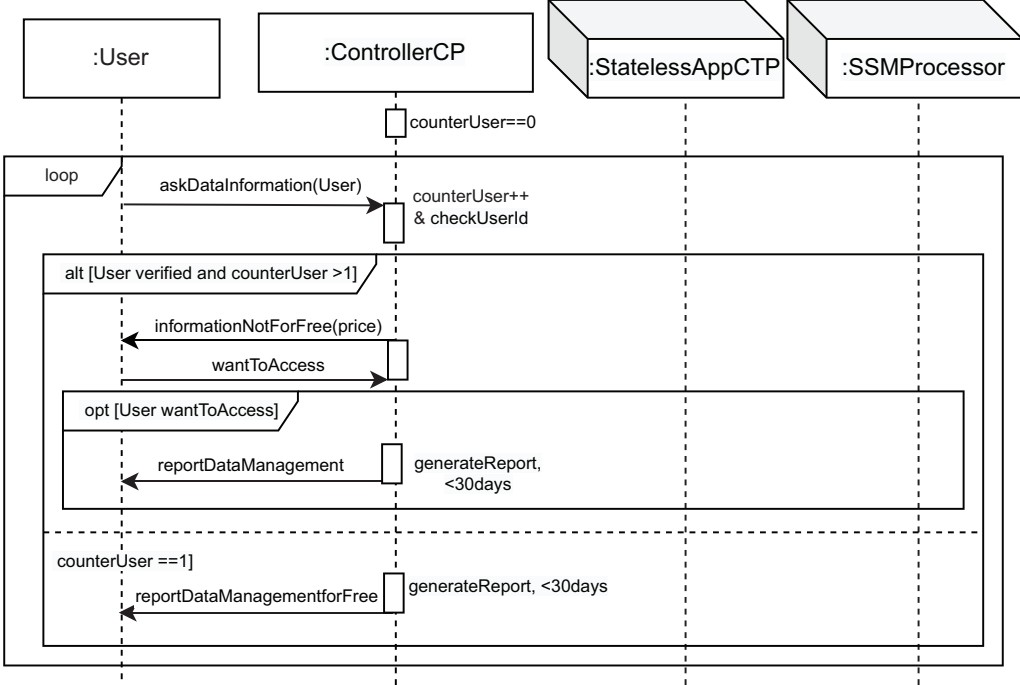

**Figure 6 InfoDataUser SD: user asks about the management of its data.**

*(EU GDPR), 2016*)). This contract defines the fields of the sticky policy for the data, that is, the permissions, the owner, the purpose, the controller, and the accessHistory, with this last field being empty at the beginning.

The controller and the processor then also sign a Service Level Agreement (*SLA* message), which allows the specific storage machine to be set as the data processor and thus process personal data on behalf of the controller. This contract defines the maximum time during which data is stored on a machine and the third parties that can access data using the *processingDuration*, and *recipients*, respectively (see the class diagram in Appendix A). Hence, data will be stored and processed on that machine. These contracts are defined according to Article 13.2 GDPR (*General Data Protection Regulation (EU GDPR), 2016*). The GDPR specifies that the processing of data by a processor shall be governed by a contract (Article 28.3, Recital 44 and 109 GDPR (*General Data Protection Regulation (EU GDPR), 2016*)), where the processing period (*maxTime* parameter) is established, which is based on the defined *processingDuration*. After that, the user can transfer its data to the controller (*sendData* message).

Note that the associated sticky policy is a property of the data (see the class diagram in Appendix A), and from that moment on, the controller oversees and is responsible for controlling the processing of the data. The *saving controller in SP* action allows the controller to save its identity in the sticky policy by using the *controller* property. Then, considering Article 13.1 GDPR (*General Data Protection Regulation (EU GDPR), 2016*) ("the controller shall provide the data subject with some information"), the controller informs the user (*info* message) about the period for which data will be stored (*maxTime*), the third parties *recipients*, the identity, and the contact details of the controller (*ControllerCP*). Article 14 GDPR (*General Data Protection Regulation (EU GDPR), 2016*) defines what the controller must do when personal data has not been obtained from the data subject (for instance, from a different company). In this scenario, the data controller must inform the user and provide the same guarantees as before. Therefore, once the user is informed, the controller can store the user data on the storage machine (*sendData* message).

Subsequently, the controller enters a loop to handle the messages received in the system until the time to store the data expires (*time ≤ maxTime* condition) or the user orders the deletion of its data. For this purpose, a *loop* combined fragment is used to model the repetition of the interactions within it. Note that this combined fragment is inside a sequence diagram fragment called *sd main*. This is an *interaction use* in UML and allows us to reference it from other diagrams by simply using the label *ref* together with the name of the fragment (*e.g.*, *ref main* in this case). Therefore, note that the *sd main* combined fragment can also end when the deletion of the data is ordered by the user, after which the data controller orders the processor to erase the data subject data and any of its copies (message *eraseData*), in accordance with Article 28.3.g GDPR (*General Data Protection Regulation (EU GDPR), 2016*). If this happens, the processor acts by *removing Data & copies* (at the bottom of this main SD-diagram) and the controller acts by *removing SP in Log* and removing the corresponding data from the log (self-message *removeDatainLog*).

The *alt* combined fragment inside the *loop* allows us to model the occurrence of different events that can occur in the system. The first event (first part of the *alt* combined fragment) occurs when the user wants to add new restrictions to its data policy (*the user wants to add new restrictions* condition). This event allows the user (data subject), for

instance, to withdraw consent to third parties at any moment, that is, to change their access permissions in accordance with Article 7.3 GDPR (*General Data Protection Regulation (EU GDPR), 2016*). The message *newRestrictions* containing its data (*Data*) and the new restrictions (*newSP*) is set. The *newSP* parameter is of type *StickyPolicy*, and describes a list of third parties with their associated permissions (*permission* field) as an array of elements of type *PermissionPerTP*, which are *(S, I)* pairs, where both *S* and *I* are a list of lists of TPs (Third Parties, defined by *StatelessAppCTP*), and *S* defines who is authorized to grant permissions for data access, and *I* the third parties with writing permission over the data (see "Combination and Data Aggregation" and Appendix A). The controller then saves these new constraints to the log (*saving new SP in log* action). Therefore, these new data restrictions must be updated on all the machines that store the data. To do this, the *update* message is sent to the *SSMProcessor* roles with the *Data*, *machines*, and *newSP* parameters, to specify the data, the machines where these are stored, and the new sticky policy, respectively.

The second event (second part of the *alt* combined fragment) corresponds to the user's right to rectify inaccurate personal data *via* the data controller without undue delay, Article 16 GDPR (*General Data Protection Regulation (EU GDPR), 2016*) (*the User wants to rectify Data* condition). The user sends the message *rectifyData* to the *ControllerCP*, with two parameters, namely *Data* and *newData*, corresponding to the old and new data, respectively. The data must be updated on all the machines where it is stored. To do this, a new message is sent to all the storage machines (*SSMProcessor*) that contain the data to inform them about the data rectification (message *newData*), with these three parameters: *Data*, *machines*, and *newData*.

Article 17 GDPR (*General Data Protection Regulation (EU GDPR), 2016*) regulates the user's right to delete its data (*the User wants to remove Data* condition of the *alt*). Therefore, the third event (third part of the *alt* combined fragment) occurs when the user orders the removal of its data, sending the message *removeData*, which contains the data (*Data*) to be removed. After that, the controller searches for all the possible machines in the log that store the data to erase them (*Seaching Data Ubications* action) and updates the log by deleting all entries with the deleted information (represented by the recursive self-message *removeSPinLog*). After that, an *eraseData* message is sent to the corresponding processors, with the *Data* and *machines* parameters, to indicate the data to be deleted and the *SSMProcessors* that store it, respectively. Note that these interactions are within a *break* combined fragment, which allows us to model that once the data has been eliminated, the execution leaves the loop[7]. Let us remark that the user is the only entity authorized to eliminate its data, so we do not consider a special type of permission for this purpose.

Another possible event occurs when a StatelessAppCTP, that is, a third-party (TP), wants to access the user's data (*the TP wants to access Data* condition; fourth part of the *alt* combined fragment). As Fig. 3 shows, the *interaction use* called *accessing_Data* is executed. This interaction use shows the implementation of the interactions between the roles from the system to access the user's data (see Fig. 4). As can be seen in this figure, the TP that wanted to access the data must request the user's consent by sending a *consent* message to the controller, as per Article 6 GDPR (*General Data Protection Regulation (EU GDPR), 2016*).

---

[7] As explained above, the execution of the loop can end for two reasons: the time for storing the data has elapsed or the user's data has been deleted at the request of the user.

The parameters of the *consent* message are the data to be accessed (*Data*), the purpose for which the TP wants to access the user's data (*purpose*), and the action to be performed on them (*action*, of *ActionType* type, see Fig. 4). In response to this message, the controller requests the TP to identify itself with the *askAuthentification* message. Then, the TP sends its identity in the *tp* message. Once the controller receives the identification of the TP, it verifies its permissions in the SP associated with this data. If the TP has no permission to access the data, the controller sends a *consent* message to the user. Now, the user has the right to object (Article 21 GDPR (*General Data Protection Regulation (EU GDPR), 2016*)). This behavior is represented in the second part of the *alt [User consents]* and equates to doing nothing. This situation also covers Article 22 GDPR (*General Data Protection Regulation (EU GDPR), 2016*) (that is, the user has the right not to be subject to decisions coming only from automatization—including profiling). Otherwise, the condition *[User consents]* is fulfilled, and the *ok* message is sent with three parameters: *tp*, *purpose* and *ActionType*, corresponding to the identification of the TP, the purpose for which the data is accessed, and the type of permission to access, respectively. Then, the controller updates its log, adding the information about this new access (*adding access information in AccessLog* action), including a new record on it, and sends the permission to the TP (*permission* message), in accordance with Article 7 GDPR (*General Data Protection Regulation (EU GDPR), 2016*). Therefore, the *loop* combined fragment is executed if the condition *[tp has permission or User consents]* is fulfilled, that is, the TP had permission or the user has accepted. This structure is used to model the repetition of TP operations (*read*, *write*, and *combine*) on the specified data.

TP operations are of *ActionType* type (see the figures in Appendix A), *i.e.*, the TP can read data (*readData* message) or write data (*writeData* message) (see Fig. 4). However, the TP can also combine several sets of read data. Then, the corresponding part of the *alt* combined fragment will be executed, depending on the action that the TP wishes to perform:

- If the TP wants to read, the first part of *alt* (*[TP wants to read and has permission for reading] condition)* is executed. If the TP wants to read, and since it has obtained consent, then the *readData* message is sent to the SSMProcessor to read the data.
- If the TP wants to write, the second part of *alt* is executed (*[TP wants to write and has permission for writing]* condition). In this case, the *writeData* message, which has the new data content as parameter (*content* parameter), is sent from the TP to the *SSMProcessor* to write the data and it allows the TP to overwrite the data with that content. The *maxTime* is 0 since the storage time is unchanged.
- Finally, if TP wants to combine several data, the third part of *alt* is executed (*[TP wants to combine and has permission for reading and writing]* condition). In this case, the *alt [purpose==statistical]* allows us to model the two different behaviors depending on the purpose of access.

1. If the purpose contributes to statistics on customers or the population (statistical purpose), the first part of *alt* (condition *[purpose==statistical]*) is executed. In this

8 We assume that the TP has previously read the other data.

case, a *readData* message is used to read the data. Then, the TP acts by *calculating aggregated newData and newSP in trustedM*. This action allows the TP to perform a statistical operation on the data[8], which are being held on a trusted machine dedicated to this purpose in our cloud architecture. Later, a *writeData* message allows the TP to write the new data in the storage (*newData* parameter for *maxTime* period, which is aggregated data). In this case, the new data is owned by the TP, which makes decisions on it.

2. If the purpose is not statistical (for instance, tax returns), once the TP has read the data (*readData* message), it acts by *combining Data and calculating newData and newSP*. This action is run on a non-reliable cloud machine. Later, a *writeData* message allows the TP to write the new data (*newData* parameter for *maxTime* period of time). Finally, the interaction use *main* is executed to manage the newly generated data.

"Combination and Data Aggregation" explains in detail how the new SP is generated (*calculating newSP*) when data aggregation or combination of the data is performed.

Then, the *accessHistory* field of the new SP is adapted to include the data access information (*updating accessHistory of SP* action). The controller acts by *updating SP in log* to modify the corresponding SP in the log, in the *accessHistory* field. Finally, in accordance with Article 12 GDPR, the user is informed of the data accessed through the message *accessinformation*.

The following event in Fig. 3 occurs when the controller (*ControlerCP*) wants to know about changes to the user's data, modeled by the fifth part of the *alt* structure (*the ControllerCP wants to subscribe to notification of breaches* condition; Article 34 GDPR). Thus, the controller subscribes to receive notifications when data breaches occur. For this purpose, the interaction use (*subscription ref* frame) is executed.

Figure 5 shows this interaction use. The controller can subscribe to notification of any changes detected by the processor through the use of the *subscribe* message, which allows control of data changes at any time. This message has four parameters: *Data*, the machines the controller wants to control (*machines* parameter), the violations detected (*breaches* parameter), and the maximum subscription time (*MaxSubscriptionTime* parameter). This maximum time is set by the GDPR at 72 h, in accordance with Article 33, in which the data controller must notify the violation of personal data to the competent supervisory authority in accordance with Article 55, unless it is unlikely that the violation of personal data poses a risk to rights and freedoms of data subjects. The *breaches* parameter is an array of *(Data, TP, actionType, newData, newlocation)*. In the event of a breach (*alt [breaches]*), the controller receives a notification message (*notify* message). This message has three parameters: *Data*, *machines* and *breaches*. Subsequently, the controller checks whether changes to the data have been logged in the log (*check changes* action), and has 72 h to inform the user, represented by the *informBreaches* message. If during the maximum subscription time (*MaxSubscriptionTime*) any changes or breaches are not detected, the *timeoutSuscription* message is sent to the controller from the processor. This process allows the controller to audit any changes that occur by verifying the information included in its log.

Cambronero et al.
2024
10.7717/peerj-cs.1898

The last event occurs when the user requests information about the handling of its data, in accordance with Article 15 GDPR. This corresponds to the last part of the alt in Fig. 3 (*the User wants information about the data management* condition). In this case, the interaction use called *infoDataUser* is executed (see Fig. 6). According to the GDPR, the first time the user requests information about the processing of its data, it will be provided free of charge. However, if further copies are requested, a reasonable fee reflecting administrative costs should be required. We model this behavior as follows. First, a counter named *counterUser* is defined and initialized to zero. Then, a *loop* structure is included to model the possible repetitive behavior of the user when requesting this information. The message *askDataInformation* from the user to the controller models this request. The controller then performs the *counterUser++ & checkUserId* action to increase the value of *counterUser* and search for this user's information. Subsequently, the *alt* combined fragment with the *User verified and counterUser > 1* condition allows the execution of the first part to inform the user that they have to pay a fee, represented by the *informationNotForFree* message with the parameter *price*. Therefore, the user can decide whether to pay and receive the information (*wantToAccess* message). In this case, the *opt [User wantToAccess]* is executed and the controller generates the report in less than 30 days (*generateReport, <30 days* action), and sends the *reportDataManagement* message to the user. However, if this is the first time the user has requested the information (*counterUser == 1* condition), the controller generates the report within those 30 days and sends the information for free, with the *reportDataManagementforFree* message to the user.

## Combination and data aggregation

At this point, we provide details about the rules applied in the data combination operation, which are inspired by the ideas presented in *Stefan et al. (2012)*. Permissions are DC labels: tuples of the form $\langle S, I \rangle$, where $S$ and $I$ are conjunctive normal forms on entities without negative literals. $S$ represents entities whose permission is required to grant access to the data, while $I$ represents the entities that have full access to the data. DC labels have a *can-flow-to* relation $\sqsubseteq$ defined as:

$$\frac{S_1 \rightarrow S_2, \; I_1 \rightarrow I_2}{\langle S_1, I_1 \rangle \sqsubseteq \langle S_2, I_2 \rangle}$$

There are two operations defined on DC labels:

- $\langle S_1, I_1 \rangle \sqcup \langle S_2, I_2 \rangle = \langle S_1 \wedge S_2, I_1 \vee I_2 \rangle$
- $\langle S_1, I_1 \rangle \sqcap \langle S_2, I_2 \rangle = \langle S_1 \vee S_2, I_1 \wedge I_2 \rangle$

If we consider $\mathscr{D}$ to be the set of DC labels, then the pair $(\mathscr{D}, \sqsubseteq)$ forms a lattice:

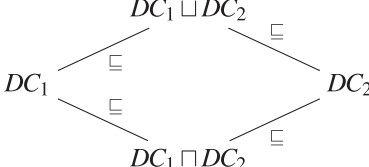

when combining two DC labels, for instance $DC_1$ and $DC_2$, we must keep the less restrictive DC label stronger than both: $SP_1 \sqcup SP_2$. For the sake of simplicity, we consider the combination of only two different data. However, it can easily be extended to combine more data. Then, we can define the combination operator of two DC labels $c : \mathscr{D} \times \mathscr{D} \mapsto \mathscr{D}$, defined as $c(DC_1, DC_2) = SP_1 \sqcup SP_2$. We obtain $DC_1 \sqsubseteq c(DC_1, DC_2)$ and $DC_2 \sqsubseteq c(DC_1, DC_2)$. Let us illustrate this with an example.

**Example 5.1.** Suppose *DS1* has all the data to complete her tax form. *DS1*'s data has also included the data for her husband, *DS2*. Hence, the sticky policy for the data is the following:

$SP_{tax}$={
  {permission: $\langle DS1 \wedge DS2, DS1 \vee DS2 \rangle$},
  {owner: $DS1 \wedge DS2$},
  {purpose: taxes},
  {controller: cloud provider},
  {accessHistory: $H$}
}

*DS1* has a tax agent *GestF* who prepares the tax form. Since *GestF* needs to access *DS1*'s data, this agent must request access to read the data and then create a new document combining *DS1*'s data and its own data. The resulting sticky policy is:

$SP_{tax}$={
  {permission: $\langle DS1 \wedge DS2, DS1 \vee DS2 \rangle$},
  {owner: $DS1 \wedge DS2$},
  {purpose: taxes},
  {controller: cloud provider},
  {accessHistory: $H \cup [(GestF, taxes, read)]$}
}

The resulting tax form has the following sticky policy

$SP_{taxform}$={
  {permission: $\langle DS1 \wedge DS2 \wedge GestF,$
   $DS1 \vee DS2 \vee GestF \rangle$},
  {owner: $DS1 \wedge DS2 \wedge GestF$},
  {purpose: taxes},
  {controller: cloud provider},
  {accessHistory: []}
}

However, when aggregating data, the aggregating entity must request permission from all the entities required by the DC label of each aggregated set of data. The entity then creates new data owned by the entity, aggregating the data. The historical field of the aggregated data should reflect this access. For instance:

**Example 5.2.** Suppose that *SB* (Santander Bank) wants to average the taxes paid by their clients. There are two clients, *DS1* and *DS2*, whose tax data have the following sticky policy:

$SP_{DS1}$={
  {permission: $\langle DS1 \wedge DS2 \vee GestF \rangle$},

```
    {owner: DS1 ∧ GestF},
    {purpose: taxes},
    {controller: cloud provider},
    {accessHistory: H₁}
}
```
$SP_{DS2}$={
```
    {permission: ⟨DS1,DS2⟩},
    {owner: DS2},
    {purpose: taxes},
    {controller: cloud provider},
    {accessHistory: H₂}
}
```

Thus *SB* needs to ask *DS1* and *GestF* for permission to access *DS1*'s tax form and only *DS2* for *DS2*'s tax form. The resulting sticky policies are:

$SP_{DS1}$ = {
```
    {permission: ⟨DS1 ∧ GestF,DS1 ∨ GestF⟩},
    {owner: DS1 ∧ GestF},
    {purpose: taxes, satistical}, \
    {controller: cloud provider},
    {accessHistory: H₁ ∪[(SB,statistical,read)]}
}
```
$SP_{DS2}$ = {
```
    {permission: ⟨DS2,DS2⟩},
    {owner: DS2},
    {purpose: taxes, statistical},
    {controller: cloud provider},
    {accessHistory: H₂ ∪[(SB, statistical, read)]}
}
```

We can observe that *SB* has read *DS1*'s and *DS2*'s data for statistical purposes. The sticky policy of the aggregated data (the average) is:

$SP_{avg}$={
```
    {permission: ⟨SB,SB⟩},
    {owner: SB},
    {purpose: statistical},
    {controller: cloud provider},
    {accessHistory: []}
}
```

## Architectural model

For simplicity, in this section, we only present a summary of the model that defines the proposed cloud infrastructure. The complete detailed description is available in Appendix A. In previous works (*Bernal et al., 2019*; *Cambronero et al., 2021*), we have presented some aspects of the cloud infrastructure, but without considering data storage

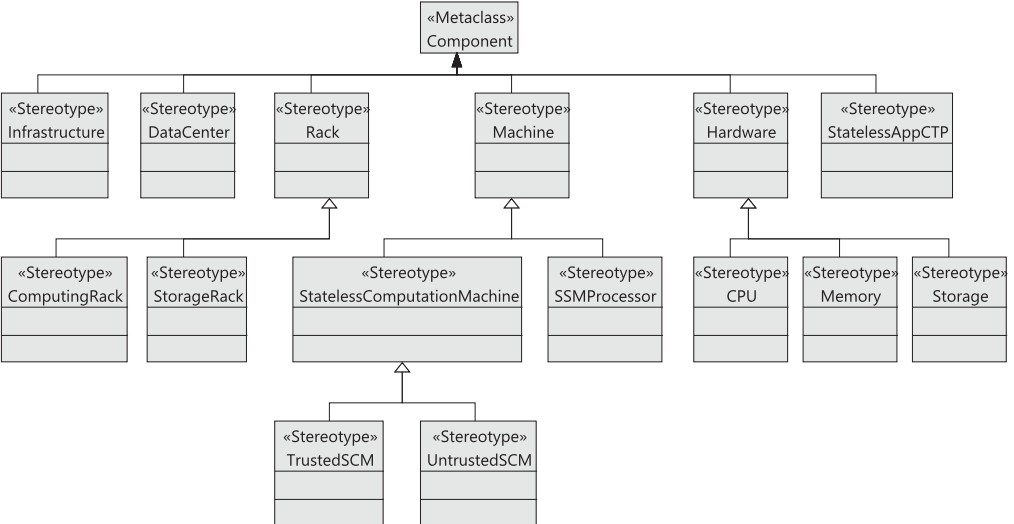

**Figure 7 Model4_DataCTrack profile: cloud-GDPR infrastructure stereotypes.**

and management. In contrast, in this work, we focus on this aspect of the cloud, defining a GDPR-compliant architecture to manage the data of users who access the cloud. Hence, our architecture provides data privacy management, GDPR compliance, and data tracking. In particular, data privacy management and data tracking are provided through the use of sticky policies (see "Sticky Policies"). Similarly, GDPR compliance is validated using OCL rules (see "Validation of the Model4 DataCTrack Models").

Figure 7 shows the stereotypes defined to model the main components of the cloud infrastructure. First, by extending the *Component* metaclass, the *Infrastructure* stereotype represents the cloud infrastructure together with the complementary services offered by the cloud provider. The stereotypes *DataCenter*, *Rack*, *Machine*, *Hardware*, and *StatelessAppCTP* also extend the *Component* metaclass. In this way, the cloud infrastructure consists of a set of data centers, which in turn are composed of a collection of racks (*Rack*). A rack belongs to two subtypes (*StorageRack* or *ComputingRack*), depending on the type of machine (*Machine* stereotype) it contains. In particular, the machine can be a stateless storage machine (*SSMProcessor* stereotype) or a stateless computing machine (*StatelessComputationMachine* stereotype), respectively. Therefore, a storage rack will be made up of several (*storage*) machines, and a computing rack by one or more computing machines. Note that a computing machine has two subtypes: it can either be a *TrustedSCM* or an *UntrustedSCM*. Trusted machines are served by controllers in our cloud architecture with the special purpose of statistical use (defined in "Running Example") and store read-only data, whereas untrusted machines are accessible to anyone and can be used for other purposes, such as taxes or insurance calculation. The *Hardware* stereotype represents the components that any machine will have, and has three sub-stereotypes: *CPU*, *Memory*, and *Storage*. Finally, the *StatelessAppCTP* stereotype represents third-party applications seeking to access the data.

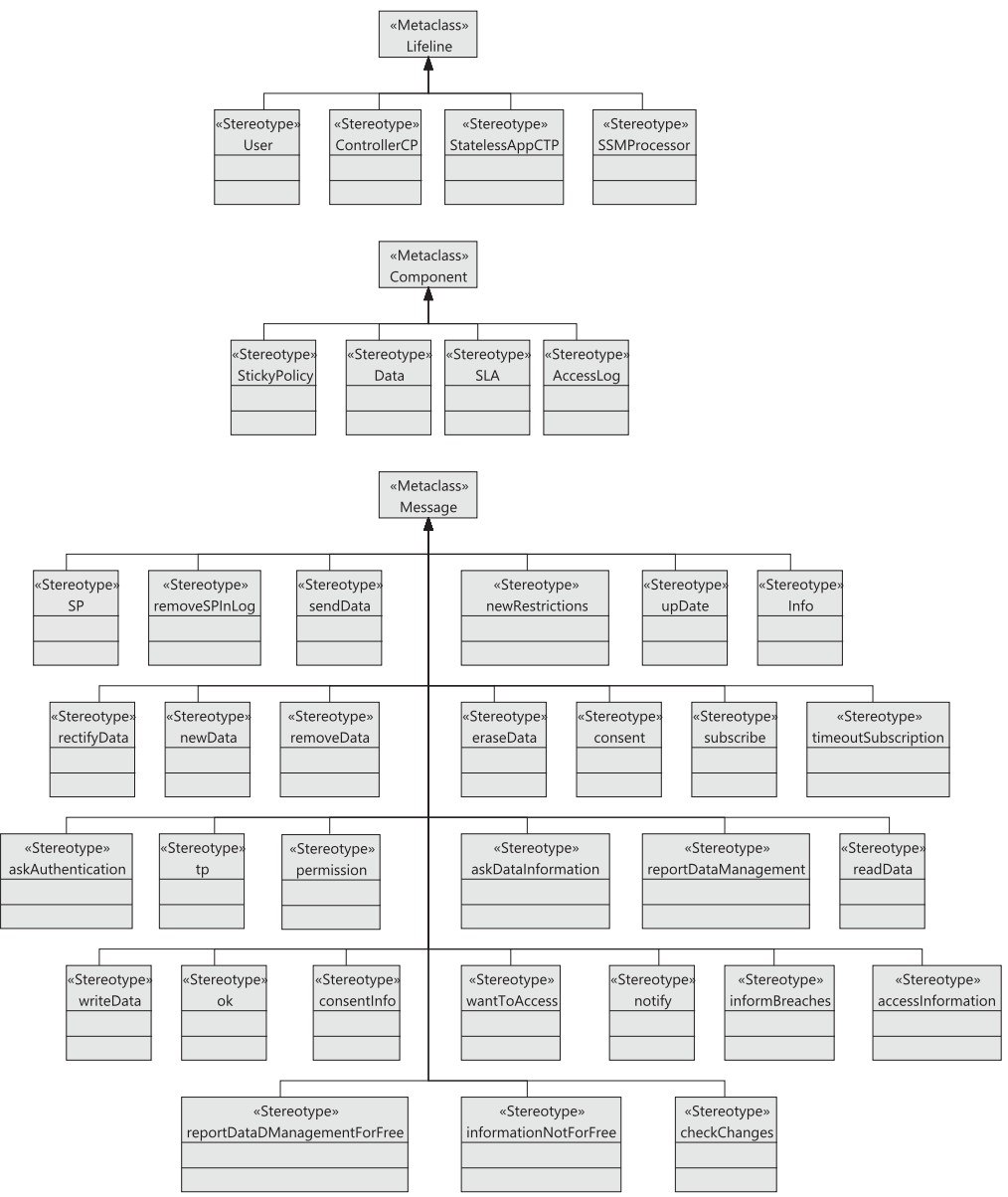

**Figure 8 Model4_DataCTrack profile: interaction stereotypes.**

As mentioned, the cloud infrastructure is made up of several data centers, many of which have similar or identical configurations, as they are typically purchased in bulk. For this reason, the relationships between components are defined as associations between stereotypes. In Appendix A, these associations are illustrated graphically; they are also discussed in detail.

The definition of the stereotypes used for the interaction (Interaction Model) appears in Fig. 8. The *User*, *ControllerCP*, *StatelessAppCTP*, and *SSMProcessor* stereotypes extend the *Lifeline* metaclass. Also, there are the roles that interact in the cloud architecture (see Figs. 3–6). The stereotype *User* represents the data subject or user. The cloud service provider

**Table 3 Controller log _Accesslog_ structure.**

| TP | L1 | SP1 | O | Action | NL | NSP |
|----|------|-----|-----------|------------|---------|-----|
| tp | Storage | SP | PList[1..*] | ActionType | Storage | SP |

(*ControllerCP*) represents the user's data controller, as explained earlier. The different third-party applications that access the data make up the *StatelessAppCTP* stereotype. And finally, the machines (*SSMProcessor* stereotype) represent the entities which will storage and process of the data, thereby becoming the data processors. Next, the *StickyPolicy*, *Data*, *SLA*, and *AccessLog* stereotypes extend the metaclass *Component*. These stereotypes represent the system components used for data representation and control. Finally, all the messages exchanged in the interaction extend the *Message* metaclass.

Appendix A shows the attributes and relationships between interaction stereotypes as associations of stereotypes. In this appendix, these attributes and relationships are fully described.

We should point out that the controller uses the log (*AccessLog* stereotype) to store information about all the data accesses and changes to the SP associated with the data, which is made up of several fields of different types, as described in Appendix A. Table 3 summarizes this controller log structure. As can be seen, for each data access the following information is stored in a log record: *TP*, the third party accessing the data, of type *StatelessAppCTP, tp* for short; *L1*, the initial data **L**ocation (storage machine) of type *Storage*; *SP1*, the initial data **S**ticky **P**olicy, of type *Sticky Policy*; *O*, the list of entities (third parties or users) granting permission to access the data, of type *PList*; *Action*, the action performed on the data, of type *ActionType*; *NL*, the **N**ew **L**ocation of the data, of type *Storage*; and finally, *NSP*, of *Sticky Policy* type, which stores the **N**ew **S**ticky **P**olicy, in case of changes to the initial sticky policy.

# VALIDATION AND THREAT MODEL

This section first outlines the procedure for validating the models generated using our tool and then describes the threat model of our approach.

## Validation of the Model4_DataCTrack models

To facilitate the validation process, we have established a set of OCL rules (*Object Management Group (OMG), 2014*), which can be found in Appendix B, encompassing the complete collection of OCL rules.

These rules have been categorized into two distinct groups. The first group, known as the *structural* rules, primarily focuses on the conventional relationships between stereotypes and their corresponding properties. Appendix A provides a comprehensive description of these constraints. Table 4 presents the most noteworthy examples.

**Rule STR-1** validates that the set of data included in an instance of the *upDate* message (*self.data*) is present in all machines to which the message is destined. This is accomplished by verifying that the list of data sets stored in each machine (*m.storage.*

**Table 4 Subset of OCL Rules defined for structural consistency.**

| Attributes | Value |
|---|---|
| **Rule STR-1** | **all_machines_must_contain_data_to_update** |
| Severity | ERROR |
| Context | upDate |
| Specification | `self.machines->forAll(m \| m.storage.data->` `includes(self.data))` |
| **Rule STR-2** | **newLocation_machine_must_be_under_sla_with_controller** |
| Severity | ERROR |
| Context | ControllerCP |
| Specification | `self.accesslog ->` `forAll(log \| self.sla->` `exists(sla \| log.newLocation.sla` `-> includes(sla)))` |
| **Rule STR-3** | **no_empty_rectify_fields** |
| Severity | ERROR |
| Context | rectifyData |
| Specification | `self.newData->forAll(f \| f.value.size() > 0)` |

*data*) includes the data from the message, for all machines in the destination machines list of this message (*self.machines*).

**Rule STR-2** checks whether the new processor mentioned in any *AccessLog* of the controller, where the data has been copied to, is also under the Service Level Agreement (SLA) with the controller of this data. To achieve this, the controller's log list (*self.accesslog*) is examined to validate the existence of an SLA in the controller's SLA list (*self.sla*) that is included in the SLA list of the *newLocation* machine in the log (*log.newLocation.sla*).

**Rule STR-3** ensures that the data introduced in any instance of a *rectify* message does not violate the data accuracy principle of the General Data Protection Regulation (GDPR) by containing empty fields. For this purpose, it is verified that all fields in the *newData* attribute of the message (*self.newData*) have a size (number of characters in the string) greater than 0.

The second group comprises rules that pertain to the specific restrictions imposed by the GDPR. Given the significance of these rules in the context of this article, we will now give a more detailed explanation of the rules that we consider most relevant. Refer to Table 5 for a summary of these rules.

**Rule GDPR-1** verifies that every machine in the list of machines to which an *update* message is intended has been assessed as compliant with GDPR standards by an authoritative GDPR entity. In other words, it ensures that the *GDPRCompliance* attribute for all these machines is set to *true*.

**Rule GDPR-2** validates that within the *accessHistory* list of a given *StickyPolicy*, none of the recorded accesses have an associated purpose that is not included in the allowed set of purposes specified by the *purpose* attribute of that policy.

**Table 5 Subset of OCL rules derived from GDPR.**

| Attributes | Value |
|---|---|
| **Rule GDPR-1** | **upDate_destinantion_machines_comply_with_GDPR** |
| Severity | ERROR |
| Context | upDate |
| Specification | `self.machines -> forAll(m | m.GDPRCompliance=true)` |
| **Rule GDPR-2** | **allowed_access_purpose** |
| Severity | ERROR |
| Context | StickyPolicy |
| Specification | `self.accessHistory->`<br>`    forAll(his | his.purpose->`<br>`     forAll(p | self.purpose->includes(p)))` |
| **Rule GDPR-3** | **tp_in_history_given_permissions** |
| Severity | ERROR |
| Context | AccessLog |
| Specification | `self.accessHistory ->`<br>`    forAll( his | AccessLog.allInstances ->`<br>`     exists( log | log.tp = his.tp`<br>`      and log.action = his.actionPerformed))` |
| **Rule GDPR-4** | **log_access_match_sp_access** |
| Severity | ERROR |
| Context | AccessLog |
| Specification | `AccessLog.allInstances() ->`<br>`    forAll(log | log.sp.accessHistory ->`<br>`     exists(access | access.tp = log.tp and`<br>`      access.actionPerformed=log.action))` |
| **Rule GDPR-5** | **no_access_permission_given_without_user_consent** |
| Severity | ERROR |
| Context | permission |
| Specification | `permission.allInstances() ->`<br>`    forAll(ok.allInstances() ->`<br>`     exists(okmsg|self.purpose ->`<br>`      forAll(p | okmsg.purpose -> includes(p)) and`<br>`       okmsg.permissionType = self.permissionType) and`<br>`        consentInfo.allInstances() ->`<br>`         exists(consentmsg | self.purpose ->`<br>`          forAll( p | consentmsg.purpose->includes(p)) and`<br>`           consentmsg.action = self.permissionType and`<br>`            consentmsg.tp = StatelessAppCTP.allInstances()->`<br>`             select(tp | tp.base_Lifeline.coveredBy ->`<br>`              includes(self.base\_Message.receiveEvent))))` |

**Rule GDPR-3** ensures that all third parties listed in the *accessHistory* field of a *StickyPolicy* possess the appropriate permissions defined for them within the *permission* list of that policy. Specifically, it examines the *I* field of the sticky policy's *permission* field.

**Rule GDPR-4** raises an error if a specific data access recorded in the controller's log (*AccessLog*) does not have the corresponding access included in the *accessHistory* of the

associated *StickyPolicy*. This rule examines the *accessHistory* list of the *StickyPolicy* to verify whether the access has been included.

**Rule GDPR-5** checks that a third party cannot obtain permission to access the data without obtaining prior consent from the corresponding data subjects. This implies that the preceding *consentInfo* and *ok* messages have been sent with the same purpose and permission.

## Threat model

In this section, we describe the threat model of a system to which our profile applies to provide a basis for understanding the potential risks and the corresponding safeguards to ensure the security of the described system.

Adversaries may attempt to gain illicit access to user data stored in the cloud. In this sense, strong authentication mechanisms must be employed, along with the need to obtain explicit consent from data subjects each time access is requested. Another critical threat involves data manipulation, where adversaries seek to manipulate user data within cloud infrastructure. Implementing strict controls and obtaining consent from data subjects whenever data is subject to modification can help overcome this threat.

Unauthorized disclosure of sensitive user information to third parties caused by privacy breaches is another possible threat. Mitigation strategies include strict compliance with GDPR guidelines, implementing sticky policies for fine-grained access control, and encrypting sensitive data to safeguard privacy. In this regard, policy abuse is another privacy threat, requiring regular policy reviews, enforcement of access control, and continuous monitoring of policy violations. Likewise, inadequate logging and monitoring practices make it difficult to detect and respond to security incidents. Comprehensive logging mechanisms, real-time monitoring tools, and the establishment of incident response procedures are needed to adequately address security incidents.

Unauthorized access through compromised third parties is another concern. This can be overcome by regularly auditing third-party entities and their permissions, along with enforcing strict access controls. Finally, inadequate management of user consent can lead to unauthorized data processing. Implementing robust consent mechanisms, regular updates to consent preferences, and ensuring compliance with GDPR guidelines can help overcome this issue.

## THE MDCT TOOL

This section presents the computer-aided design tool that supports our framework, making it easy to use our modeling framework. The tool, which has the same name as the framework, focuses on the modeling of cloud systems and supports Model4_DataCTrack for the management of sensitive data in the context of GDPR. MDCT has been developed by extending Papyrus UML (*Lanusse et al., 2009*), which is an Eclipse-based graphical editing tool for UML2. MDCT contains a modeling part, in which the UML profile can be used to define a specific GDPR-compliant cloud architecture, as defined in "Modeling Data Tracking in Cloud Systems". For this purpose, the graphical interface provides all of the stereotypes and data types used for the proposed infrastructure and interaction (as shown

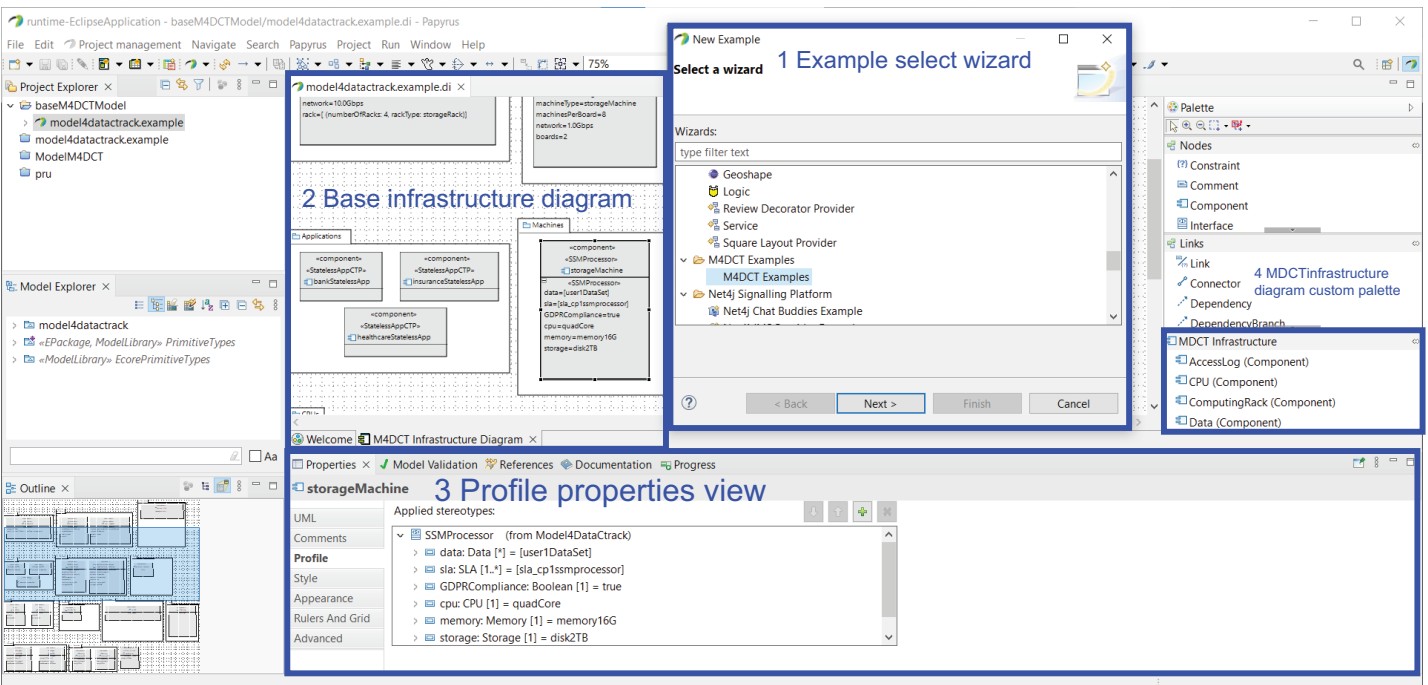

**Figure 9** Tool interface and base MDCT example opening wizard.

in Figs. 7, 8, and the figures in Appendix A), allowing the tool user (cloud providers) to provide different values for some of its parameters. All these elements are available through custom palettes to make it easy to design MDCT models with just drag and drop. MDCT also implements the validation of the model restrictions set using OCL rules, as detailed in "Validation and Threat Model". Finally, our tool includes an example, in which the infrastructure and the interaction of a basic cloud architecture are modeled. This example can be loaded and extended to avoid starting from scratch. It is available at https://zenodo.org/doi/10.5281/zenodo.10380128.

Figure 9 displays a screenshot of the graphical interface of the MDCT tool, featuring four highlighted sections. The first section is dedicated to the *selection of a wizard*, allowing users to choose from example models of a GDPR-compliant architecture. Users can also access these examples through the main menu by selecting File ❯ New ❯ Example . This wizard enables users to load predefined profile models instead of designing them from scratch. Once the models are loaded, they are presented in the diagram editor (box 2 in Fig. 9), showcasing the cloud infrastructure diagram. In this diagram, users can select any element and modify its attributes using the profile tab in the Properties view (box 3). Additionally, users can easily add new elements by dragging them into the diagram through the customized M4DCT diagram palette (box 4). This process can also be executed by incorporating the appropriate component, lifeline, or message and applying the desired stereotype in the profile tab of the Properties view .

In addition, Fig. 10 displays a screenshot illustrating how data tracking is managed in the MDCT tool. This example is derived from the running example presented in "Running

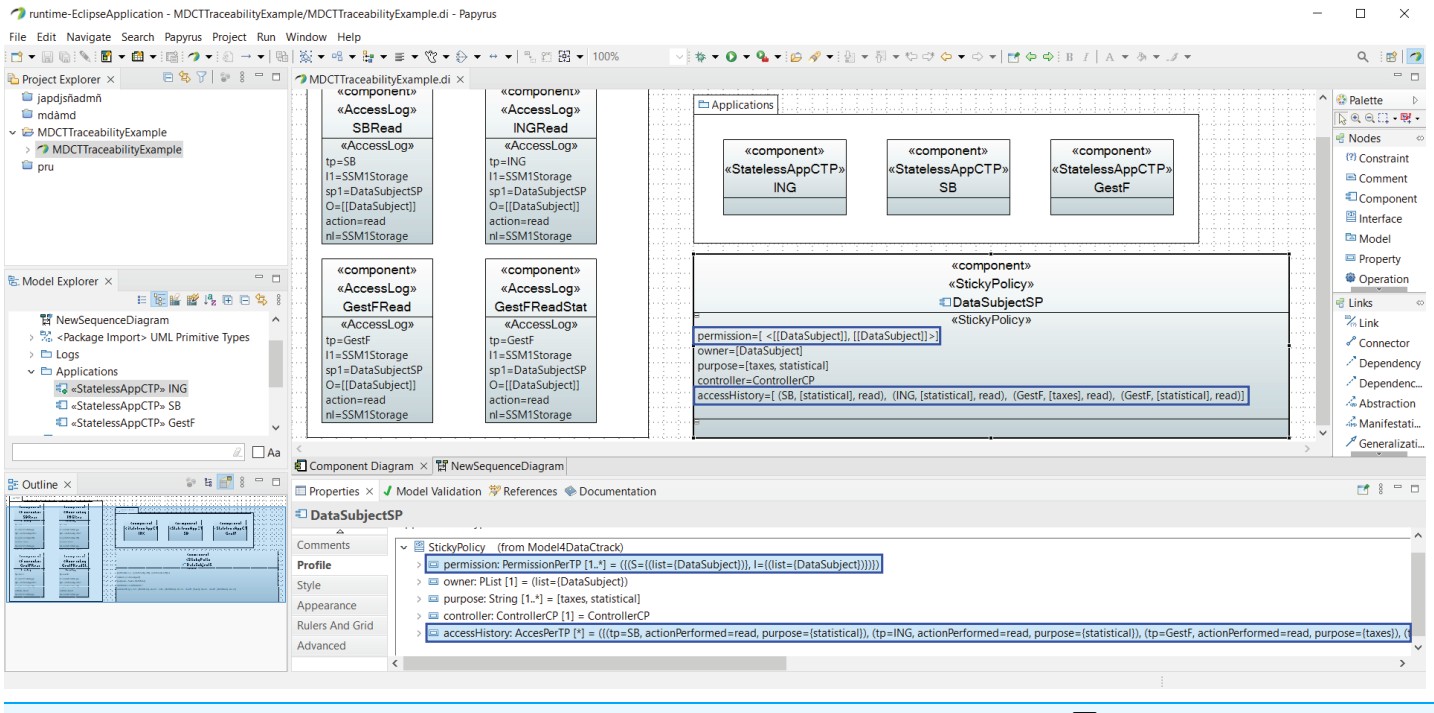

**Figure 10 A brief example of how traceability is portrayed in MDCT.**

Example". In this instance, the initial sticky policy (*accessHistory: [(SB, statistical, read), (ING, statistical, read), (GestF, taxes, read), (GestF, statistical, read)]*) indicates that *SB* and *ING* read the data for statistical purposes, followed by *GestF* performing read access for both tax and statistical purposes. This entire process is recorded in the *accessHistory* field, enabling the tracking of data as it contains comprehensive information about all data access instances.

The *accessHistory* field of the sticky policy allows tracking of all data accesses. In this case, all these *read* accesses have been saved to it, as depicted in the *DataSubjectSP accessHistory* field at the bottom of Fig. 10. Notably, these third parties must have permission to execute these accesses, a requirement checked by using Rule 6 (see Table 5). The *accessHistory* field of data in the sticky policy provides information about the third party's read permission. In this case, the *permission* field has the same value for *S* and *I*, specifically ⟨*DS*, *DS*⟩, signifying that only DS (the data subject) can give consent to access the data and has permission to write.

## DISCUSSION

In this section, we discuss the main considerations in our framework and the threats to its validity.

### Considerations

Below is a list of some important considerations concerning our framework that we would like to highlight:

1) **Support.** The support of two experts in the GDPR has allowed us to design and develop our modeling framework.

2) **Types of machines according to the purpose of data access.** In our cloud architecture, we consider that the cloud provider offers two types of machines to process the data, depending on the purpose for which the data is accessed. The two types of purposes we consider are statistical and non-statistical. When the access is for statistical purposes, the processing is carried out on trusted machines, and the processor in charge of the treatment will be the owner of the new data. In this case, only third parties to whom the owner authorizes access to the data may access the data. These trusted machines are read-only, and when several of these data are combined for that statistical purpose, the new data will be aggregated data. However, when the purpose of access is not statistical, the processing is carried out on unreliable machines, and if several data sources are combined, the owners of the combined data will be the owners of the original data while the access permissions will be the most restrictive (see "Combination and Data Aggregation").

3) **EU or non-EU members.** We propose a controlled cloud architecture in which the cloud provider works with machines that may or may not be in members of the EU, but all of them ensure an adequate level of protection according to the GDPR, Article 45. For this purpose, we have included the *GDPRCompliance* field in the *SSMProcessor* stereotype (see Appendix A). The value of this field is checked by OCL to ensure that machines acting as processors are GDPR compliant.

4) **Consent.** In our architecture, when a third party wants to access the data and does not have permission to do so, the user's consent must be requested to authorize such access, as can be seen in Fig. 4. In this case, if the user consents to access, this third party will be included in the list of permissions on that data (indicating the type of permission granted) and will thus have access to the data.

5) **Supervisory authorities.** In this article, we do not explicitly model the supervisory authority as a role in the system as we consider it to be an element outside our cloud architecture. However, interactions with this supervisory authority are easy to include.

## Validity threats

- **Internal validity.** A potential threat to internal validity is that we have interpreted the text of the GDPR provisions to create a cloud architecture. However, this is recommended for any company that operates in the cloud, whether inside or outside the EU, when these are companies that offer goods or services to people in the EU. In our case, this phase was carried out in collaboration with people with a good knowledge of the field (the authors of this work, who are experts in the GDPR) to minimize the threat posed by such a subjective interpretation. Of course, we cannot rule out subjectivity, but we do provide our interpretation accurately and explicitly. Furthermore, our model is publicly available.

- **External validity.** Our framework focuses on defining and validating a GDPR-compliant cloud architecture, which has been designed with input from legal experts in data protection. Therefore, this allows us a certain degree of confidence in the generalization of our results. However, future studies exemplifying our model in different cloud domains with their corresponding legal aspects will be critical in deciding the completeness and applicability of our framework in real-world scenarios.

The validation process allows us to verify inappropriate access or breaches of customer data confidentiality. Thus, we can conclude that certain recommendations be given to the entity responsible for data security (the controller) to define its architecture in the cloud. In this case, the data controller is the cloud provider, who is responsible for the data of the cloud's customers and for third party access.

## CONCLUSIONS AND FUTURE WORK

This article introduces the MDCT computer-aided design framework. This framework is made up of a UML profile as a means to model and validate a GDPR-compliant cloud architecture (which is recommended for cloud providers offering services in the EU), a set of OCL rules to validate the models, and a Papyrus-based tool. The UML profile introduces the cloud infrastructure and the interactions between the different roles in the context of the GDPR. The profile models key GDPR considerations such as user consent/withdrawal, the purpose of access, and data transparency and auditing. In addition, it also considers data privacy and data tracking. Data privacy is included through sticky policies associated with the data, allowing us to define data permissions, the data owner, the controller, and the purpose.

In this work, we have considered the purpose of access to be statistical or non-statistical. The cloud provider offers trusted machines to process the data in the case of statistical purposes. Thus, various data can be added to a new set of data, whose owner will be the entity that performs the data aggregation, and its permissions will be decided by the owner. For other purposes, the data processing takes place on non-reliable machines, and the combination of data generates new data, whose owners are the owners of all the individual data, and the permissions of the sticky policy are the most restrictive. Data tracking is made possible by adding a new field to the sticky policy associated with the data, which allows us to record which third parties access the data and for what purpose. Furthermore, our framework allows us to model complex cloud scenarios, representing the underlying cloud infrastructure and the third parties that access the data. It also incorporates OCL rules to validate important restrictions and features in accordance with the GDPR, data privacy, and data tracking.

For future work, we have several lines of research planned. We intend to enrich the profile by including other GDPR features, such as interaction with supervisory authorities. We also intend to translate our models into real cloud infrastructures, such as Amazon Web Services or Microsoft Azure. For this purpose, we pretend to use some novel technologies, such as Infrastructure as Code (*Artac et al., 2017*). Furthermore, we plan to broaden the spectrum of possible cloud configurations by considering different hardware

configurations and not just using different types of physical machines, depending on the purpose of data access.

### Funding
This work was supported by the Spanish Ministry of Science and Innovation (co-financed by European Union FEDER funds) projects "FAME (Metodologías Avanzadas para Arquitecturas, Diseño y Pruebas de Sistemas Software)", reference PID2021-122215NB-C32; and the Region of Madrid (grants FORTE-CM, S2018/TCS-4314 and PR65/19-22452). The research of Ricardo J. Rodríguez was supported by the Aragonese Government under Programa de Proyectos Estratégicos de Grupos de Investigación (DisCo research group, ref. T21-23R). The funders had no role in study design, data collection and analysis, decision to publish, or preparation of the manuscript.

### Grant Disclosures
The following grant information was disclosed by the authors:
Spanish Ministry of Science and Innovation (co-financed by European Union FEDER funds) Projects "FAME (Metodologías Avanzadas para Arquitecturas, Diseño y Pruebas de Sistemas Software)": PID2021-122215NB-C32.
Region of Madrid: FORTE-CM, S2018/TCS-4314 and PR65/19-22452.
Aragonese Government under Programa de Proyectos Estratégicos de Grupos de Investigación: T21-23R.

### Competing Interests
M. Emilia Cambronero is an Academic Editor for PeerJ.

### Author Contributions
- M. Emilia Cambronero conceived and designed the experiments, prepared figures and/or tables, authored or reviewed drafts of the article, and approved the final draft.
- Miguel A. Martínez conceived and designed the experiments, performed the experiments, performed the computation work, authored or reviewed drafts of the article, and approved the final draft.
- Luis Llana performed the computation work, prepared figures and/or tables, authored or reviewed drafts of the article, theorical development, and approved the final draft.
- Ricardo J. Rodríguez analyzed the data, prepared figures and/or tables, authored or reviewed drafts of the article, and approved the final draft.
- Alejandro Russo analyzed the data, prepared figures and/or tables, authored or reviewed drafts of the article, theoretical development, and approved the final draft.

### Data Availability
  The latest version of the source code of the tool is available at GitHub and Zenodo:
  - https://github.com/uclm-es-Model4-DataCTrack/MDCT_Tool.

- Miguel Ángel Martínez Atienza. (2023). LuisLlana/MDCT_Tool: peerj (peerj). Zenodo. https://doi.org/10.5281/zenodo.10380129.

## Supplemental Information

Supplemental information for this article can be found online at http://dx.doi.org/10.7717/peerj-cs.1898#supplemental-information.

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
