# Peer review of "Towards a GDPR-compliant cloud architecture with data privacy controlled through sticky policies"

_PeerJ Computer Science, doi:10.7717/peerj-cs.1898_

## Round 0.1 · original submission · Major Revisions

There are several concerns in terms of how information is presented in the paper, which needs to be addressed. In addition to the issue of presentation, there is also a concern by one reviewer on how end-users may verify whether the desired invariants are being met. This is a very crucial point, and may require more than just clearing up the presentation aspects, but also augmenting it with more technical details and perhaps even a (sketch of) proof of security.

Reviewer 2 has suggested that you cite specific references. You are welcome to add it/them if you believe they are relevant. However, you are not required to include these citations, and if you do not include them, this will not influence my decision

Reviewer 1 ·

Basic reporting

In this manuscript, secure cloud-based architecture is proposed and considered secure data privacy and data tracking mechanisms. However, some minor issues need to be improved for acceptance of this manuscript.
1. On page 2, under the "Introduction section," the sentence - "Our proposal is based on UML and UML profiling techniques, which are well-known software development methodologies in software engineering." Please mention one or two names of the UML profiling techniques as an example for better understanding to international readers.
2. The way of mentioning the sentence is not correct using the keyword 'Note that' – "Note that we addressed the main considerations in the GDPR in order to ensure the security of user data in our proposed architecture at page 2, under "Introduction section," Please correct/remove the word from the sentence for the proper way of understanding.
3. On page 2, under the "Introduction section" and in the "Contribution section,"- the authors mentioned that – "UML profiling techniques allow us to define specific stereotypes needed to model the architecture of the cloud." It is not easy to appreciate the paper's contribution. Such type of sentence will be in the "Introduction section" only.
4. On page 6, the authors mentioned - "our architecture considers trusted stateless machines that guarantee," and on page 9 – "to create a stateless computing architecture." However, the authors should mention what a stateless machine is, what its application is, why it is called trusted, and why this proposed scheme is here. So, please incorporate such things first and then use the term for clear understanding.
5. Several abbreviations are used in this manuscript, like SB and ING, but no "Appendix" or "Keyword" section is maintained. Moreover, for the keyword 'ING,' no detail term is referred to throughout the whole manuscript, whereas SB is used on page 6, the detail term is referred to on page 18. So, those keywords are maintained in a very unprofessional manner and very difficult to understand for the learners.

6. On page 4, the authors wrote - “A UML-SD consists of a group of objects or roles that interact in a system, which are represented by lifelines”. It is not clear what is 'lifelines' and why it is used. Please clarify it with one line to make it a clear idea for learners from different fields.

7. 9. On page number 19 under subsection- "5.3 Architectural Model", the paper states that - "The complete detailed description can be found in A." the same thing is repeated on page 20 - "which is made up of which several fields of different types, as described in Figure 12, in A.", What is A here? Please make it clear.

8. On page 20, the paper states that – "L1, the initial data location (storage machine) of type Storage; SP1, the initial data sticky policy of type Sticky Policy, SP for short; O, the list of entities (third parties or users) granting permission to access the data, of type PList; Action, the action performed on the data, of type ActionType; NL, the new location of the data, of type Storage; and finally, NSP, of Sticky Policy type, which stores the new sticky policy in the case of changes to the initial sticky policy".
Please maintain an Abbreviation or Keyword section to maintain such events so that such terms should be understandable to international readers. There are better ways to represent such things.

Experimental design

9. On page 3, under the "Introduction section" and the last section of research contribution-
The authors mentioned - "Once the user of the profile (that is, the cloud provider) has parameterized the models, they are validated using OCL rules to check their correctness, according to certain restrictions and desirable characteristics.”
10. Again, it takes work to appreciate the paper's contribution. From this sentence, it seems that the models are doing their activities, but there is no contribution from the author's end. So, either remove it or correctly configure it so that it looks like the author's own contribution.

Validity of the findings

11. On page 19, the paper states – "Hence, our architecture provides data privacy management, GDPR compliance, and data tracking."But it should be added how this architecture provides such features with this architecture in 1-2 sentences here.

Additional comments

12. The authors have written "related work," but it should be under the Introduction section as a "Literature Review" or "Related Works" section, but I think it is not in the correct position. Moreover, most of the cited kinds of literature are also old; only one is found in 2023.
So, please correct the section and insert some recent references.

Reviewer 2 ·

Basic reporting

This work uses model-driven engineering approach to define UML profile to enable a GDPR-compliant cloud architecture which can achieve data privacy controlled through sticky policies described by OCL.

The main problem is that the proposed approach seems to rely on the cloud provider to apply the sticky policies for controlling and managing the third party’s access to the protected data. However, cloud provider itself is often considered as “honest but curious”, which means the cloud provider should not be fully trusted. In this paper’s case, how could the data subject know the cloud provider strictly executes the policies? Also, is there any measure used by the data owner to prevent the ‘malicious’ cloud provider? Any privacy preserving approach is applied before data outsourcing? Taking a step back, a threat model should be detailed in advance.


A table that lists all the definitions of the used notations are highly recommended.

Several related work should be cited:
David W. Chadwick, Wenjun Fan, Gianpiero Costantino, Rogério De Lemos, Francesco Di Cerbo, Ian Herwono, Mirko Manea, Paolo Mori, Ali Sajjad, and Xiao-Si Wang. "A cloud-edge based data security architecture for sharing and analysing cyber threat information." Future generation computer systems 102 (2020): 710-722.
Wenjun Fan, Joanna Ziembicka, Rogério de Lemos, David Chadwick, Francesco Di Cerbo, Ali Sajjad, Xiao-Si Wang, and Ian Herwono. "Enabling privacy-preserving sharing of cyber threat information in the cloud." In 2019 6th IEEE International Conference on Cyber Security and Cloud Computing (CSCloud)/2019 5th IEEE International Conference on Edge Computing and Scalable Cloud (EdgeCom), pp. 74-80. IEEE, 2019.

Experimental design

This paper seems to lack an implementation, e.g., applying this approach to a real cloud environment, and the paper should also present a number of experimental results based on the implementation.

Validity of the findings

Figure 9 and 10 as screenshots are not very encouraged to use, while the main problem is the fontsize is too small to be used in a case study.

---

## Round 0.2 · Minor Revisions

There are a few very minor editorial issues that have been highlighted by a reviewer. Those need to be addressed.

Reviewer 1 ·

Basic reporting

The authors have corrected the previous comments provided earlier. Moreover, the authors also have tried to configure the manuscript appropriately.
However, some additional findings (as mentioned in section 4, 'Additional comments' ) are found, and the manuscript can only be accepted after successfully fulfill all such comments.

Experimental design

The authors have corrected the previous comments provided earlier.

Validity of the findings

The authors have corrected the previous comments provided earlier. Moreover, the authors also have tried to configure the manuscript appropriately.
However, some additional findings (as mentioned in section 4, 'Additional comments' ) are found, and the manuscript can only be accepted after successfully fulfill all such comments.

Additional comments

1. At line number 89 of page number 3 of the modified manuscript, the sentence needs to be constructed correctly. It does not have any meaning – “ This tool is publicly available and its nosource code has been released under the GNU/GPLv3 license”.

2. At page number 4, line number 130, the sentence needs to be correctly constructed. Please construct the sentence in a proper format for clear understanding and with correct meaning.
“In order to facilitate the reader’s reading, a list of acronyms is provided in section ”.

3. Another point is the structure of this paper. You have mentioned under ‘Section 8’ that a list of a list ‘of acronyms’ is maintained. However, I am not getting it under section 8; instead, it is at the beginning of the manuscript. So, please check the structure of the paper then and adequately mention all such events properly.

4. In sub-section 1.1.1, Modeling and Validation of GDPR and Data Privacy under literature review on page number 4 and line number 136, you have mentioned the previous works from 2015 to 2023. But the representation is incorrect, as you have mentioned the recent work (2023) before and the old work (2015) at the end. There are better ways of representing the previous works. Instead, you must first mention the old related work and its limitations and sequentially describe the recent works later.

5. The ordering sequence as per the year is also incorrect for the sub-section “1.1.3, Data Tracking and 3GDPR” under the literature review at page 6 and line number 238.

6. Please check thoroughly the construction of the sentences properly, mainly in the introduction, contribution, related works, experimental results, and conclusion as well before submitting this manuscript.

---

## Round 0.3 · accepted · Accept

The revisions are satisfactory.